# Loss of *Jag1* cooperates with oncogenic *Kras* to induce pancreatic cystic neoplasms

Wen-Cheng Chung[1], Lavanya Challagundla[2], Yunyun Zhou[2], Min Li[3], Azeddine Atfi[4] , Keli Xu[1,5]

**Notch signaling exerts both oncogenic and tumor-suppressive functions in the pancreas. In this study, deletion of *Jag1* in conjunction with oncogenic *Kras*[G12D] expression in the mouse pancreas induced rapid development of acinar-to-ductal metaplasia and early stage pancreatic intraepithelial neoplasm; however, culminating in cystic neoplasms rather than ductal adenocarcinoma. Most cystic lesions in these mice were reminiscent of serous cystic neoplasm, and the rest resembled intraductal papillary mucinous neoplasm. Jag1 expression was lost or decreased in cystic lesions but retained in adenocarcinoma in these mice, so was the expression of Sox9. In pancreatic cancer patients, *JAG1* expression is higher in cancerous tissue, and high *JAG1* is associated with poor overall survival. Expression of *SOX9* is correlated with *JAG1*, and high *SOX9* is also associated with poor survival. Mechanistically, Jag1 regulates expression of Lkb1, a tumor suppressor involved in the development of pancreatic cystic neoplasm. Collectively, Jag1 can act as a tumor suppressor in the pancreas by delaying precursor lesions, whereas loss of Jag1 promoted a phenotypic switch from malignant carcinoma to benign cystic lesions.**

## Introduction

Pancreatic ductal adenocarcinoma (PDAC) remains one of the most lethal malignancies. Investigation of the PDAC precursors, their cellular origins, initiation and progression is of great importance for early detection and intervention for the disease. Three distinct precursor lesions of PDAC have been identified, including pancreatic intraepithelial neoplasm (PanIN), intraductal papillary mucinous neoplasm (IPMN), and mucinous cystic neoplasm (MCN). PanINs are the most important and common precursors of PDAC. They are microscopic pancreatic lesions associated with ducts, but interestingly, they can originate from acinar cells ([1]). IPMNs belong to the heterogeneous group of pancreatic cystic neoplasms. Human IPMNs display distinctive intraductal growth, and evidence from mouse models suggests that IPMNs most likely arise from the progenitor niche of the pancreatic ductal epithelium ([1]). MCNs are the most infrequent precursor lesions of PDAC and are characterized by the presence of progesterone receptor (PR)- and estrogen receptor (ER)-positive ovarian type stroma. A wide variety of genetic changes have been identified in precursor lesions with increased frequency in advanced cases, including mutations of *KRAS*, *p16INK4A/CDKN2A*, *SMAD4*, and *TP53* genes, which are largely overlapping in PanINs and IPMNs, and detected less frequently in MCNs ([2]). The mechanisms underlying the differentiation of different precursor lesions remain to be elucidated.

Notch signaling plays important yet complex roles in PDAC. Whereas Notch2 was shown to be required for progression of PanIN to PDAC ([3]), loss- and gain-of-functions of Notch1 both rendered acinar cells more susceptible to Kras-induced PanIN formation and progression ([4], [5], [6]). We recently discovered a PDAC-suppressive function for Lfng, a glycosyltransferase that modifies Notch receptors to enhance Delta ligand-mediated Notch activation and inhibit Jagged ligand-mediated Notch signaling, in the *Kras*[LSL-G12D/+];*Pdx1-Cre* mouse model, in which Kras activation started in the embryonic pancreas ([7]). Deletion of Lfng caused sustained Notch3 activation ([7]), and Notch3 was associated with Jag1 expression in human PDACs ([8]), suggesting that Lfng may inhibit Jag1-Notch3 signaling to suppress PDAC development.

Here we deleted *Jag1* in *Kras*[LSL-G12D/+];*Pdx1-Cre* to determine its role in Kras-driven pancreatic tumorigenesis. Deletion of *Jag1* accelerated early stage lesions including acinar-to-ductal metaplasia (ADM) and low-grade PanINs. Strikingly, these precursor lesions were diverted along a differentiation pathway towards cystic neoplasms instead of PDAC. Specifically, loss of Jag1 results in the formation of non-mucinous cysts reminiscent of serous cystic neoplasms (SCNs) in humans, and less frequently, IPMN. This study revealed a Jag1-controlled phenotypic switch between PDAC and largely benign cystic neoplasms, which appeared to be associated with differential expression of Lkb1 and Sox9 in pancreatic ductal cells.

## Results

### Deletion of Jag1 accelerates Kras-driven ADM and PanIN formation

We deleted *Jag1* in the mouse pancreas starting from embryonic stage using *Pdx1-Cre*. Quantitative RT-PCR showed a robust decrease

[1]Cancer Center and Research Institute, University of Mississippi Medical Center, Jackson, MS, USA  [2]Department of Data Science, University of Mississippi Medical Center, Jackson, MS, USA  [3]Department of Surgery, University of Oklahoma Health Sciences Center, Oklahoma City, OK, USA  [4]Cellular and Molecular Pathogenesis Division, Department of Pathology, Virginia Commonwealth University, Richmond, VA, USA  [5]Department of Neurobiology and Anatomical Sciences, University of Mississippi Medical Center, Jackson, MS, USA

Correspondence: kxu@umc.edu

in the level of *Jag1* mRNA in the *Jag1*$^{flox/flox}$*;Pdx1-Cre* (hereafter referred to as *Jag1*$^{KO}$) pancreas, which was further reflected by decreased expression of the Notch target genes *Hes1* and *Hey1* (Fig 1A). Western blot analysis confirmed lower level of Jag1 protein in the pancreas of *Jag1*$^{KO}$ mice compared with the *Jag1*$^{flox/flox}$ mice (Fig 1B). Similar to a previous report (9), *Jag1* ablation in the pancreatic lineage caused gradual loss of acini and replacement with adipocytes, accompanied by dilation of ducts and ductal desmoplasia (Fig 1C). Interestingly, we also found large cysts in 3 out of 24 *Jag1*$^{KO}$ mice older than 6 mo. Next, we generated *Kras*$^{LSL-G12D/+}$*;Jag1*$^{flox/flox}$*;Pdx1-Cre* mice (hereafter referred to as KJC) to determine the impact of Jag1 loss on Kras$^{G12D}$-induced pancreatic cancer initiation and progression. KJC mice developed precancerous lesions shortly after birth, without the replacement of acinar cells by adipocytes. At 1 mo of age, the pancreas in KJC mice had in average about 67% of the acinar compartment replaced by abnormal ductal structures, whereas *Kras*$^{LSL-G12D/+}$*;Pdx1-Cre* (hereafter referred to as KC) mice showed no or very few lesions at this age (Fig 1D and E). The lesions in KJC mice were positive for cytokeratin 19 (CK19), resembling ADMs or low-grade PanINs (Fig 1F). In addition, KJC pancreas displayed extensive desmoplasia, associating with robust expression of vimentin and smooth muscle actin *α* (SMA), which were negative in the acinar compartment of KC mice (Fig 1F). It has been shown that clinical specimens of pancreatic cancer express elevated levels of connective tissue growth factor (CTGF), and this correlates with the extent and intensity of desmoplasia (10). Indeed, *CTGF* mRNA level in KJC pancreas is five to six fold higher than that in KC pancreas (Fig 1G).

ADM occurs in response to pancreatic tissue injury, and similar metaplastic changes occur when exocrine cells are isolated and cultured, accompanied by up-regulation of Notch pathway genes (11). Stimulation of Notch by Jag1 diminished the proliferation of cultured metaplastic exocrine cells, whereas inhibition of Notch signaling had an opposite effect. This effect seemed to be Hes1-independent and mainly coincided with Hey1 and Hey2 expression (11). These observations are in agreement with our result showing that deletion of Jag1 accelerated ADM formation in vivo. Coincidentally, deletion of Jag1 also caused decreased expression of *Hey1* and *Hey2*, but not that of *Hes1* (Fig 1G). Thus, Jag1-mediated Notch signaling appears to suppress ADM and PanIN formation caused by Kras$^{G12D}$ expression in the pancreas starting from developmental stage.

## A phenotypic switch from ductal adenocarcinoma to cystic neoplasms by Jag1 deletion

We examined proliferation and apoptosis of pancreatic cells by immunostaining for Ki67 and cleaved caspase 3, respectively. At 1 mo of age, KJC mice showed increased proliferation of ductal cells compared to KC mice. Interestingly, the ducts in KJC mice also contained more apoptotic cells than KC mice (Fig 2A and B). We performed histological examination throughout the pathological course of pancreatic tumorigenesis in KJC mice. These mice developed PanIN-1A and PanIN-1B lesions as early as postnatal day 18 (Fig 2C–F). Dilation of the ductal structures and development of desmoplasia also started at this time (Fig 2E). Notably, the overall morphology of the main duct remained normal, and the vast majority of PanINs were not associated with large ducts (Fig 2F), suggesting that PanINs originated from the acinar compartment, as

reported in other Kras$^{G12D}$-driven mouse models (12, 13). These lesions continued to grow, becoming cystic neoplasms by 2–3 mo of age (Fig 2M). Characterization of cystic lesions in KJC mice (see below) identified two histological subtypes: type I, reminiscent of SCN (Fig 2G and H), and type II, resembling IPMN (Fig 2I and J). At 3 mo and older, 34.5% KJC mice (n = 29) and 10.5% *Kras*$^{LSL-G12D/+}$*; Jag1*$^{flox/+}$*;Pdx1-Cre* mice (n = 19) showed distended abdomen because of large-size pancreatic cysts containing clear to straw-colored fluid (Fig 2N), highlighting the importance of the gene dosage in this SCN/IPMN phenotype. Histological examination of the pancreas in 28 KJC mice (≥3 mo) found nine cases of small cystic neoplasm (coexisted with PanIN), 14 cases (50%) of SCN or IPMN lesions, and only one case (3.6%) of PDAC (Fig 2K and L). For comparison, there were two cases (6.0%) of IPMN-like lesion (no SCN-like lesion) and up to 11 cases (33%) of PDAC in 33 KC mice older than 3 mo (Fig 2O and Table 1). Thus, deletion of Jag1 was associated with significantly increased incidence of cystic neoplasms and decreased incidence of PDAC ($\chi^2$ test, $P < 0.0001$), suggesting a switch from ductal adenocarcinoma to cystic neoplasms.

The vast majority of KJC mice did not die from pancreatic cystic neoplasms. However, almost all of them developed ulcerating facial skin lesions necessitating euthanasia (Fig S1), which precluded analysis of pancreas-related death. A few KJC mice also developed tumors near the anus. *Jag1*$^{KO}$ mice did not show any skin lesions, whereas a few KC mice showed skin lesions at variable sites (data not shown). It has been shown that Pdx1 is physiologically expressed in the adult mouse epidermis, and in vitro analysis revealed differentiation-dependent expression of Pdx1 in terminally differentiated keratinocytes (14). Although we cannot rule out the possibility of the skin phenotype being related to a diabetic syndrome, the fully penetrated skin phenotype in KJC mice suggests that loss of Jag1-mediated Notch signaling may cooperate with oncogenic Kras to induce skin carcinogenesis. Interestingly, deletion of Notch1 also increased susceptibility to Kras$^{G12D}$-induced skin carcinogenesis with *Pdx1-Cre* (14).

## Characterization of cystic neoplasms in KJC mice

We characterized two histological subtypes of cystic neoplasm in KJC mice by immunohistochemistry. As expected, duct-lining epithelial cells in both types stained positive for CK19 (Fig 3A and D). Alcian blue staining as well as Muc5AC immunostaining revealed abundant apical mucin in cystic epithelial cells of type II but not type I lesions (Fig 3B, C, E, and F). IPMN and MCN are two subtypes of human pancreatic cystic neoplasms with mucinous differentiation (15). MCNs are distinct from IPMNs by an underlying ovarian-like stroma, which often shows nuclear staining of PR and ER, as well as expression of SMA and desmin (16). Although type I lesions lack the columnar mucin-producing epithelium typically found in MCN, their ovarian-like stroma show weak to modest ER expression and is SMA-positive (Fig 3G–I), similar to the MCN-like lesions in female mice overexpressing Wnt1 and Kras$^{G12D}$, which also lacked the typical mucinous epithelium (17). The serous cysts lined by cuboidal or flattened non-mucinous epithelial cells in the type I lesion are reminiscent of SCN rather than MCN. Human MCNs are presented almost exclusively in middle-aged women (96.5% patients are females) (18), whereas SCNs in humans exhibit female predilection

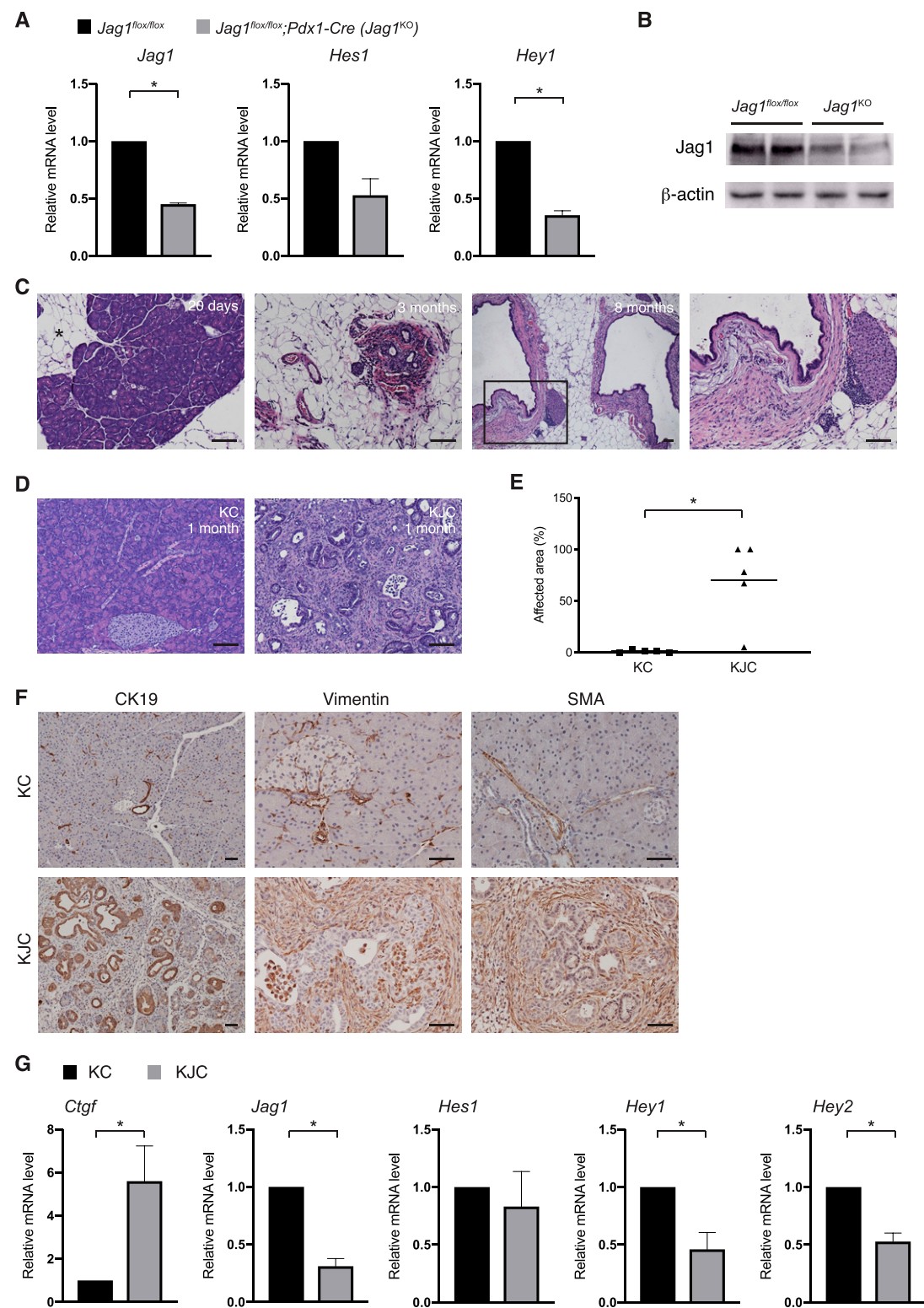

**Figure 1. Deletion of Jag1 drastically accelerated Kras-induced ADM and PanIN formation.**
**(A)** Relative mRNA levels of *Jag1*, *Hes1*, and *Hey1* in the pancreas of *Jag1^flox/flox* and *Jag1^KO* mice at postnatal day 20. **(B)** Western blot analysis for Jag1 in the pancreas of *Jag1^flox/flox* and *Jag1^KO* mice at postnatal day 20. **(C)** Representative photomicrographs of H&E–stained sections of *Jag1^KO* pancreas at the age of 20 d and 3 and 8 mo. Asterisk: area showing replacement of acinar cells with adipocytes. Note the complete loss of acinar cells, dilated ducts, and retained islets at old age. **(D)** Representative histology of the pancreas from KC and KJC mice at 1 mo of age. **(E)** Quantitation of the areas affected by ADM or PanIN lesions presented as percentage of the total area in KC and KJC mice. **(F)** Immunostaining for cytokeratin 19 (CK19), vimentin, and smooth muscle actin α (SMA) in the pancreas of 1-mo-old KC and KJC mice. **(G)** Relative mRNA levels of *Ctgf*, *Jag1*, *Hes1*, *Hey1*, and *Hey2* in KC and KJC pancreas determined by quantitative RT-PCR. *$P < 0.05$ (*t* test). **(C, D, F)** Scale bars: 50 μm in (C, D), 25 μm in (F).
Source data are available for this figure.

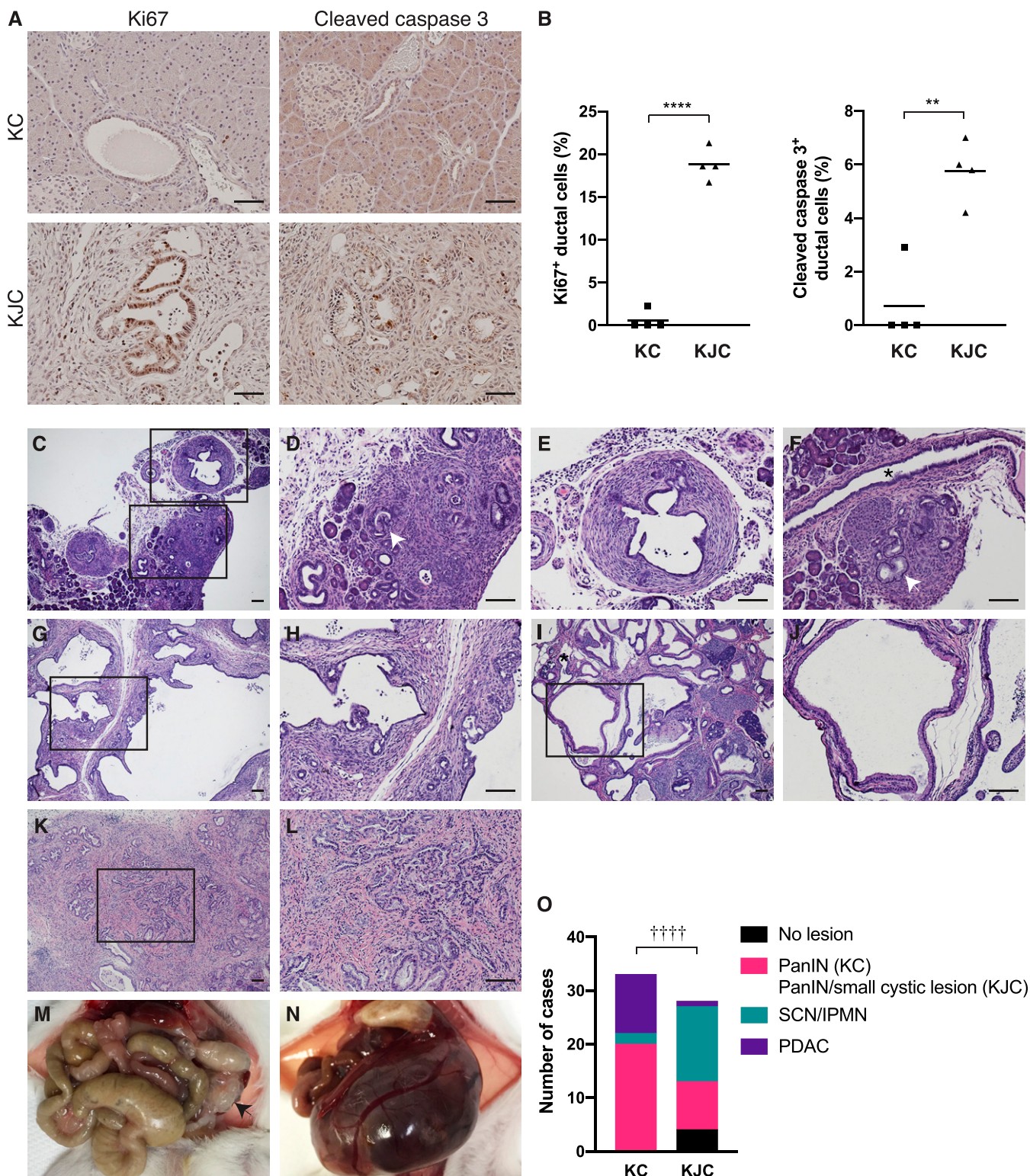

**Figure 2. Pathological course of pancreatic tumorigenesis in KJC mice.**
**(A)** Immunostaining for Ki67 and cleaved caspase 3 in the pancreas of KC and KJC mice at 1 mo of age. **(B)** Quantitation of the Ki67-positive and cleaved caspase 3–positive ductal cells presented as percentage of total ductal cells in KC and KJC mice. **(C, D, E, F)** Representative histology of KJC pancreas at postnatal day 18. White arrows: PanINs; asterisk: main pancreatic duct. **(G, H)** Representative histology of serous cystic neoplasm–like lesions in KJC mice. **(I, J)** Representative histology of intraductal papillary mucinous neoplasm–like lesions in KJC mice. **(K, L)** Rare PDAC-like histology in a 3.5-mo-old KJC mouse. **(M, N)** Gross pathologies of intraductal papillary mucinous neoplasm–like pancreas (M) and late stage serous cystic neoplasm–like pancreas (N) in KJC mice. **(O)** Quantitation of pancreatic lesions in KC and KJC mice at the age of 3 mo and older. **P < 0.01, ****P < 0.0001 ($t$ test). ††††P < 0.0001 ($\chi^2$ test). Scale bars: 25 μm in (A), 50 μm in (C, D, E, F, G, H, I, J, K, L).

**Table 1. Incidence of pancreatic lesions in KC and KJC mice.**

| | KC | | | KJC | | | |
|---|---|---|---|---|---|---|---|
| | 1–3 mo (n = 8) | 3–5 mo (n = 10) | ≥5 mo (n = 23) | ≤1 mo (n = 16) | 1–3 mo (n = 27) | 3–5 mo (n = 15) | ≥5 mo (n = 13) |
| No lesion or very few ADM/PanIN | 6 (75.0%) | 0 | 0 | 2 (12.5%) | 4 (14.8%) | 2 (13.3%) | 2 (15.4%) |
| PanIN and small cystic neoplasm | 1 (12.5%)[a] | 9 (90%)[a] | 11 (47.8%)[a] | 10 (62.5%)[b] | 11 (40.7%)[b] | 4 (26.7%)[b] | 5 (38.5%)[b] |
| SCN-like | 0 | 0 | 0 | 4 (25.0%) | 10 (37.0%) | 8 (53.3%) | 3 (23.1%) |
| IPMN-like | 0 | 0 | 2 (8.7%) | 0 | 2 (7.4%) | 0 | 3 (23.1%) |
| PDAC | 1 (12.5%) | 1 (10%) | 10 (43.5%) | 0 | 0 | 1 (6.7%) | 0 |

[a]Presence of PanIN only, no cystic neoplasm.
[b]Presence of both PanIN and cystic neoplasm.

(19). Indeed, KJC mice showed type I lesion in 15.4% (n = 39) males and 59.4% (n = 32) females ($\chi^2$ test, $P$ = 0.0001), indicating female predilection but not exclusiveness. The mucinous type II lesion does not express PR, ER, or SMA in the stroma (Fig 3J–L), therefore resembling IPMN.

Alcian blue staining in pre-weaning KJC pups showed mucinous ductal structures resembling low-grade PanINs or small-size IPMNs, as well as non-mucinous ductal lesions of relatively large size, which could be the precursor of SCN-like lesions (Fig 3M–O). To explore this phenotype further, we crossed $Rosa^{LSL-YFP}$ reporter into $Jag1^{KO}$ and KJC mice. Whereas anti-YFP immunostaining in the pancreas of $Rosa^{LSL-YFP}$; $Jag1^{flox/flox}$ mice was completely negative (Fig S2A), $Rosa^{LSL-YFP}$; $Jag1^{flox/flox}$;Pdx1-Cre pancreas showed YFP-positive cells in islets, but not in the adipocytes that have replaced acinar cells (Fig S2B and E), indicating these adipocytes are not derived from the pancreatic lineage. In $Rosa^{LSL-YFP}$;$Kras^{LSL-G12D/+}$;$Jag1^{flox/flox}$;Pdx1-Cre pancreas, all ductal lesions were YFP-positive, indicating that they have originated from Pdx1-expressing progenitors or their descendants (Fig S2C). An isotype control of the immunostaining in the same tissue was negative (Fig S2D). Interestingly, some stromal cells were YFP-positive, suggesting these cells may have undergone epithelial–mesenchymal transition (Fig S2F).

### Jag1 expression is lost in SCN but retained in PDAC

It has been shown that $Jag1$ is expressed throughout the pancreatic epithelium at embryonic day 12.5 and gradually restricted to islet clusters and ducts thereafter. By postnatal day 3, $Jag1$ expression is confined to ductal cells (9, 20). We performed Jag1 immunohistochemistry in KJC mice at various ages with different types of lesions. At 1 mo of age, KJC mice had developed extensive ADM and PanIN lesions. Although Jag1 was undetectable by immunohistochemistry in the ductal cells in wild type mice at this age (Fig 4A), Jag1 expression was up-regulated in subsets of ADM lesions (Fig 4B) and abnormal ducts (Fig 4C) in KJC mice, indicating that pancreatic deletion of $Jag1$ was incomplete in these mice. By 2–3 mo of age, many KJC mice had developed SCN-like lesions, where ductal cells lining the SCN were completely Jag1-negative, whereas blood vessels in the same section stained positive for Jag1 (Fig 4D and G). A few KJC mice formed IPMN-like lesions at this age, which exhibited no or a very low level of Jag1 expression (data not shown). Interestingly, high level Jag1 expression was observed in ductal cells of a rarely formed PDAC from an adult KJC mouse (Fig 4E and H). Thus, Jag1 expression was lost in cystic neoplasms but retained in the invasive carcinoma in KJC mice. This

could be explained by the mosaic deletion of $Jag1$ in these animals, as the $Pdx1$-Cre strain is known to display mosaic expression of Cre recombinase throughout the pancreas epithelium (21). To confirm the functionality of Jag1 expression, we performed immunostaining for Jag1 and Notch1 (or Notch2) on consecutive sections of KJC pancreas. There was no overlapping between Jag1 expression and presence of cleaved Notch1 (data not shown). However, cytoplasmic staining of Jag1 and nuclear staining of Notch2 were observed at the same location of consecutive sections (Fig 4F and I), suggesting that Jag1 may activate Notch2 during Kras-initiated pancreatic cancer development. These findings are in corroboration with a previous report showing that Notch2 functions in ductal cells and PanIN lesions and are required for PanIN progression and malignant transformation, and that deletion of Notch2 combined with Kras$^{G12D}$ expression resulted in MCN-like cystic lesions in a subset of mice (3).

We analyzed $JAG1$ expression in two published human gene expression data of matching pairs of PDAC and adjacent non-tumor tissue (22, 23). In both data sets, $JAG1$ expression is significantly higher in tumors than in non-tumor tissues (Fig 4J). Moreover, analysis using two of the human data sets containing survival information found significant association between high $JAG1$ expression and poor overall survival among pancreatic cancer patients (Fig 4K). We performed PathwayMapper (24) analysis using The Cancer Genome Atlas (TCGA) pancreatic adenocarcinoma data set hosted in the cBioPortal database. Six of the Notch signaling pathway genes, including $JAG1$, $JAG2$, $NOTCH3$, $MAML1$, $MAML2$, and $MAML3$, were up-regulated in this data set (Fig 4L). Expression heat map of these genes showed an active Jag1-Notch signature in the $JAG1$-high specimens (Fig 4M). Previous immunohistochemical study found that JAG1 expression was significantly higher in intraductal papillary mucinous carcinoma than in intraductal papillary mucinous adenoma and was significantly related to recurrence, suggesting that JAG1 levels also reflect IPMN aggressiveness (25). Thus, human patient data corroborate our findings in mice, in which we found that Jag1 expression is required for the progression of precancerous lesions to PDAC, and Jag1 levels determine the aggressiveness of those lesions.

### Loss of Sox9 expression in cystic neoplasms in KJC mice

Sox9 is a ductal fate determinant and a recent study suggests that Sox9 may prevent dedifferentiation of pancreatic ductal cells and

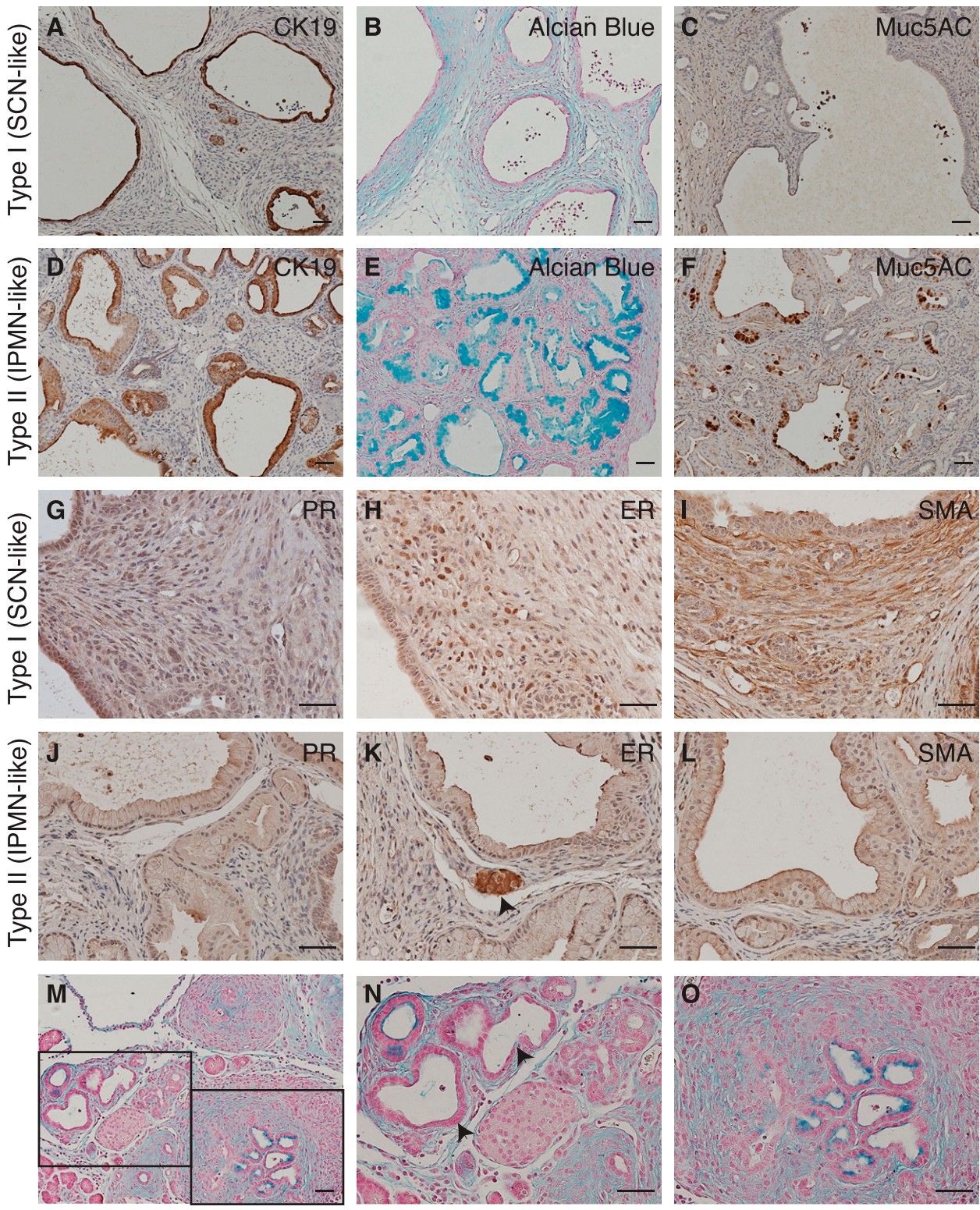

**Figure 3. Immunohistochemical characterization of serous cystic neoplasm (SCN)–like and intraductal papillary mucinous neoplasm (IPMN)–like lesions in KJC mice.**
**(A, B, C)** Anti-CK19 immunostaining (A), Alcian blue staining (B), and anti-Muc5AC immunostaining (C) in KJC pancreas with SCN-like lesions. **(D, E, F)** Anti-CK19 immunostaining (D), Alcian blue staining (E), and anti-Muc5AC immunostaining (F) in KJC pancreas with IPMN-like lesions. **(G, H, I)** Immunostaining for progesterone receptor (G), estrogen receptor (H), and smooth muscle actin α (SMA) (I) in KJC pancreas with SCN-like lesions. **(J, K, L)** Immunostaining for progesterone receptor (J), estrogen receptor (K), and SMA (L) in KJC pancreas with IPMN-like lesions. Arrow in (K): positive staining in the islet. **(M, N, O)** Alcian blue staining of KJC pancreas at postnatal day 18. Panels (N, O) are high-magnification images of the areas in panel (M). Arrows in (N): non-mucinous ductal lesion. Pancreatic tissues in panels (A, B, C, D, E, F, G, H, I, J, K, L) were harvested from KJC mice at 2–3 mo of age. Scale bars: 25 μm.

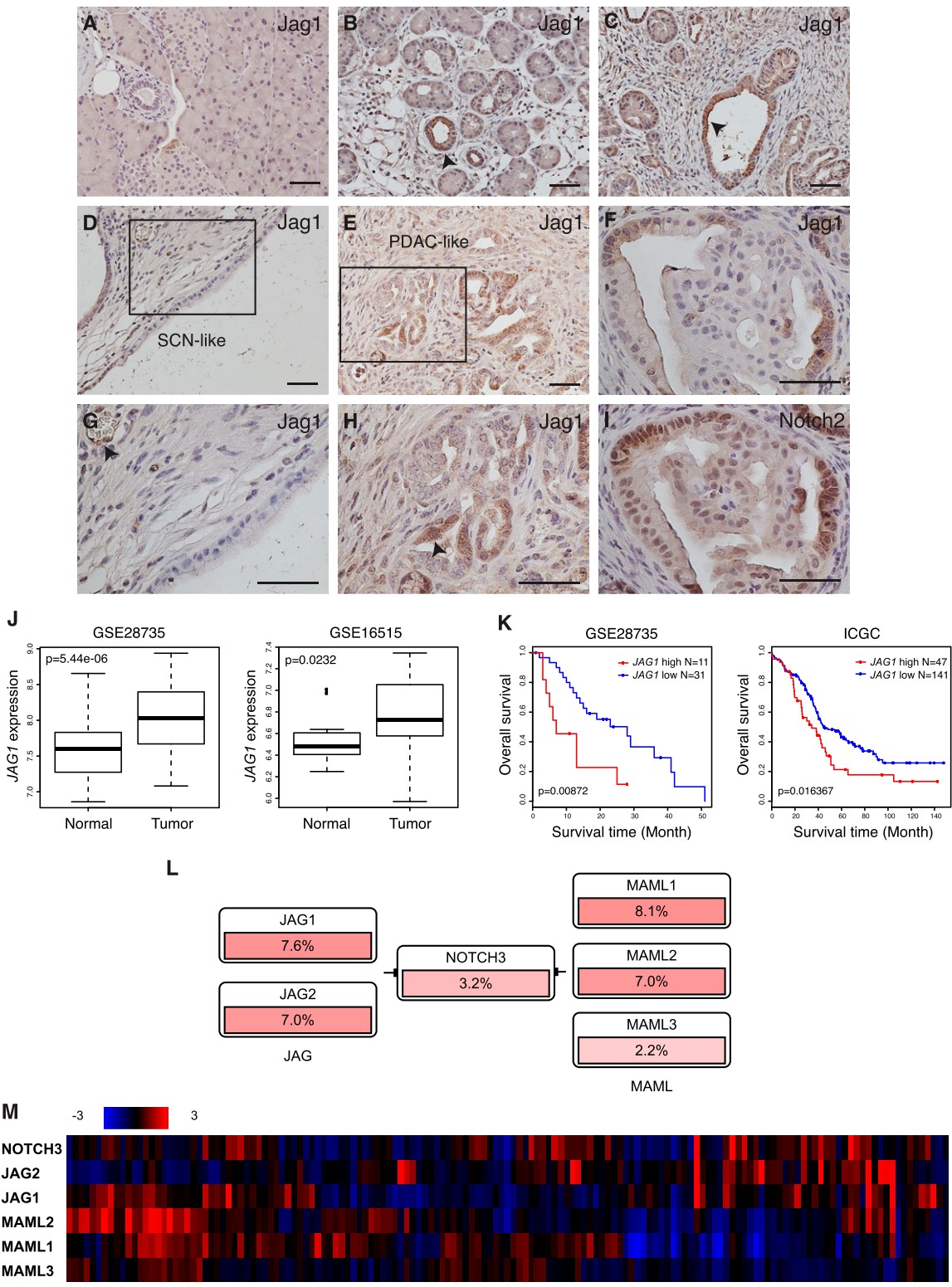

**Figure 4. Jag1 expression in KJC mice and in human pancreatic cancer patients.**
**(A)** Jag1 immunostaining in the pancreas of wild-type mice at 1 mo of age. **(B, C)** Jag1 immunostaining in the pancreas of KJC mice at 1 mo of age. **(B, C)** Arrows point to ADM lesion (in panel B) and an abnormal duct (in panel C) with elevated Jag1 expression. **(D)** Representative Jag1 immunostaining in the pancreas of adult KJC mice showing serous cystic neoplasm–like lesion. **(E)** Jag1 immunostaining in a rarely formed PDAC-like lesion in adult KJC mice. **(F)** Jag1 immunostaining in KJC pancreas at 2 mo of age. **(D, G)** High-magnification image of the area in panel (D). Arrow: Jag1 expression in the blood vessel. **(E, H)** High-magnification image of the area in panel (E). Arrow: Jag1 expression in tumor cells. **(F, I)** Immunostaining for Notch2 on adjacent tissue section of panel (F). **(J)** Mean expression values of *JAG1* in pancreatic cancer and neighboring non-tumor tissues from two human patient data sets. **(K)** Overall survival analysis in pancreatic cancer patients with high or low *JAG1* expression using Kaplan–Meier methods. **(L)** Genes of the Notch signaling pathway showing elevated expression in the The Cancer Genome Atlas pancreatic adenocarcinoma data set. **(M)** Expression heat map of the selected genes in the The Cancer Genome Atlas pancreatic adenocarcinoma data set. Scale bars: 25 μm.

IPMN formation (26). Interestingly, Sox9 is a direct target of Jag1-mediated Notch signaling in the biliary system (27, 28), and is regulated by Hes1 in the pancreas during pancreatic tumorigenesis (29). It is conceivable, therefore, that Sox9 expression may be altered in the Jag1-deficient pancreas. Indeed, immunostaining in KJC mice found that ductal cells lining SCN-like lesions were completely negative for Sox9, and IPMN-like lesions showed no or very weak Sox9 staining, whereas a rarely formed PDAC showed clear nuclear staining of Sox9 in ductal tumor cells (Fig 5A–C). Likewise, Notch2 nuclear staining was readily detected in PDAC cells, but not in the ductal cells lining the SCN-like or IPMN-like lesions (Fig 5D–F). These data suggest that cystic neoplasms in KJC mice have lost Sox9 expression, associated with loss of Notch2 activation. Western blot analysis showed significantly lower level of Sox9 protein in KJC mice with large cystic lesions than in KC mice at 4–6 mo of age (Fig 5G), suggesting that deletion of Jag1 is effective at down-regulating Sox9 expression during Kras$^{G12D}$-driven pancreatic tumorigenesis. Interestingly, analysis in TCGA pancreatic adenocarcinoma data set found a positive correlation between *JAG1* and *SOX9* expressions (Fig 5H), and high *SOX9* mRNA level is associated with poor overall survival, whereas low *SOX9* expression is associated with significantly longer survival (Fig 5I and J).

### Jag1 regulates Lkb1 expression in pancreatic ductal cells

*LKB1* gene inactivation has been involved in human IPMNs (30), and it is more common in IPMNs than in PDACs (31). Deletion of *Lkb1* in the mouse pancreas resulted in progressive acinar cell degeneration, ADM, and development of serous cystadenomas (32). Mice with an inhibitory knock-in allele of *Lkb1* also showed cystic structures in the pancreas (33), and postnatal duct-specific deletion of *Lkb1* caused ductal dilation and ADM (34). *Lkb1* haploinsufficiency cooperated with Kras$^{G12D}$ to promote pancreatic cancer, some of which exhibits a cystic morphology (35). Moreover, inactivation of Lkb1 and expression of Kras$^{G12D}$ in pancreatic ducts synergized to induce IPMN (36). Given the similarities between these mice and KJC mice, we wondered whether Jag1 could regulate Lkb1 expression under Kras$^{G12D}$-driven pancreatic tumorigenesis conditions. Immunohistochemistry showed Lkb1 expression in a subset of PanIN ductal cells in KC mice (Fig 6A), whereas in KJC mice, Lkb1 staining is positive in very few cells of IPMN-like lesions and completely negative in SCN-like lesions (Fig 6B and C). Indeed, quantitation of Lkb1$^+$ epithelial cells showed a significant decrease in KJC mice with SCN-like lesions compared to KC mice as well as KJC mice harboring IPMN-like lesions (Fig 6D). Although statistically insignificant, quantitative RT-PCR found decreased *Lkb1* mRNA expression in the pancreas of KJC mice compared with KC mice at 2–3 mo of age (Fig 6E). We isolated primary cells from KJC mice to directly test whether Jag1 regulates Lkb1 expression in pancreatic cells. Treatment of these cells with exogenous Jag1 caused up-regulation of the Notch target genes *Hes1* and *Hey1* as well as *Lkb1* (Fig 6F). Treatment with the gamma secretase inhibitor (GSI) that inhibits Notch signaling had no effect on *Lkb1* expression despite down-regulation of *Hes1* and *Hey1* (Fig 6G). Because Jag1 was deleted in the vast majority of the pancreatic cells in KJC mice with SCN-like lesions, Notch signaling decreased by GSI in these cells is most likely Jag1-independent. The fact that GSI had no effect on Lkb1 expression, whereas treatment with Jag1 increased Lkb1 in these cells suggests that Lkb1 is regulated by Jag1-dependent

Notch signaling. To test this hypothesis, primary cells from KJC mice were treated with Jag1 together with GSI (or DMSO as control). Indeed, the presence of GSI resulted in decreased mRNA levels of *Hes1* and *Hey1*, as well as a modest but significant decrease in *Lkb1* mRNA level (Fig 6H). Next, we tested the effect of Jag1 or GSI in human PDAC cell lines. Treatment with exogenous Jag1 caused a drastic increase of *LKB1* mRNA in Miapaca2, and less significantly in Panc1 cells (Fig 6I). Conversely, incubation with GSI caused a decrease of *LKB1* mRNA in Panc1, but no effect in Miapaca2 (Fig 6J). *JAG1* expression is lower in Miapaca2 than Panc1 (https://portals.broadinstitute.org/ccle/page?gene=JAG1). This may explain why adding Jag1 caused a more significant increase of *LKB1* expression in Miapaca2 than in Panc1, whereas treatment with GSI down-regulated *LKB1* in Panc1 but not in Miapaca2. Collectively, these data suggest that Jag1 may influence pancreatic tumorigenesis in part through the regulation of Lkb1.

### Ductal cell–specific deletion of Jag1 in the adult pancreas does not lead to cystic neoplasms

Pancreatic ductal cells have been identified as cell-of-origin of IPMN (26, 37, 38). We tested whether deletion of Jag1 in conjunction with Kras$^{G12D}$ expression in the ductal cells of adult pancreas would cause IPMN. For this, we crossed *Jag1*$^{flox/flox}$ or *Kras*$^{LSL-G12D/+}$;*Jag1*$^{flox/flox}$ with *Sox9-CreER*, which express a tamoxifen-inducible Cre recombinase in the ductal lineage. For simplicity, these mice will be referred hereafter as *Jag1*$^{KO-Duct}$ and KJC$^{Duct}$ mice, respectively. We treated these mice with tamoxifen at 1 mo of age and monitored for tumor development up to 4 mo post-tamoxifen administration. Histological examination of the pancreas in five KJC$^{Duct}$ mice failed to show any IPMN or SCN lesions, and only one mouse showed a single PanIN lesion (Fig S3C and D). Noteworthily, ductal cell–specific deletion of Jag1 alone in adult mice (*Jag1*$^{KO-Duct}$) caused no morphological abnormality in the pancreas (Fig S3A and B), suggesting that loss of acini, dilation of ducts, and ductal desmoplasia seen in *Jag1*$^{KO}$ mice (Fig 1C) were likely due to deletion of Jag1 in pancreatic progenitor cells during embryonic stage. Collectively, these data reveal that deletion of Jag1 affects pancreas homeostasis and cooperates with Kras$^{G12D}$ to induce SCN or IMPN when this genetic alteration occurs during pancreatic development. Interestingly, it has been reported that epithelial cells in SCNs are ultra-structurally similar to centroacinar cells (19) and that Jag1 expression is required for the maintenance of centroacinar cells as an environmental niche in the developing pancreas (39). These findings together with our genetic studies using mouse models suggest that primitive cells in the developing pancreas may serve as cell-of-origin of cystic neoplasms, including SCNs and IPMNs. SCN-like lesions may have arisen from unidentified progenitor cells with complete loss of Jag1 and concurrent Kras activation.

## Discussion

This study showed that deletion of Jag1 accelerated Kras$^{G12D}$-mediated ADM and PanIN development, and that expression level of Jag1 dictates the progression of preneoplastic lesions in the pancreas: complete and partial loss of Jag1 resulted in SCN-like and IPMN-like cystic neoplasms, respectively, whereas retained Jag1

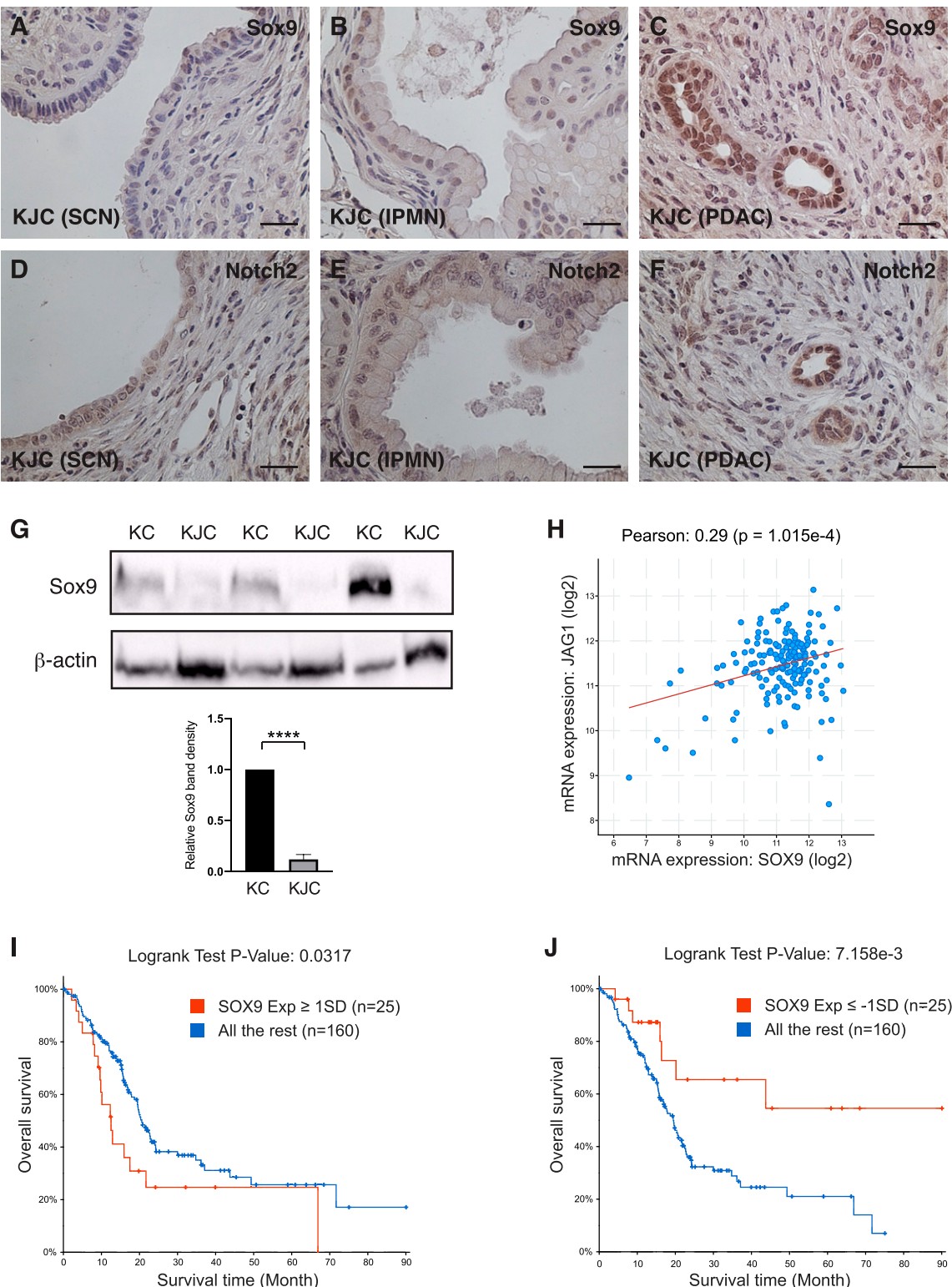

**Figure 5. Decreased Sox9 expression in KJC mice and association between *SOX9* expression and overall survival in pancreatic cancer patients.**
**(A, B, C)** Representative photomicrographs of Sox9 immunostaining in serous cystic neoplasm–like lesion (A), intraductal papillary mucinous neoplasm (B), and PDAC (C) in KJC mice. **(D, E, F)** Representative photomicrographs of Notch2 immunostaining in serous cystic neoplasm–like lesion (D), intraductal papillary mucinous neoplasm (E), and PDAC (F) of KJC mice. **(G)** Western blot analysis for Sox9 in KC and KJC mice at 4–6 mo of age. The relative levels of Sox9 normalized with the level of β-actin are quantitated and presented as fold change. **(H)** Scatterplot for *JAG1* and *SOX9* gene expressions in human pancreatic adenocarcinomas from the The Cancer Genome Atlas data set (186 samples). **(I)** Kaplan–Meier analysis of the overall survival in pancreatic cancer patients with *SOX9* expression higher than 1× SD above the mean,

expression led to the progression into PDAC. From a mechanistic perspective, Jag1 may regulate the expression of Lkb1 and Sox9, both of which play instrumental roles in Kras$^{G21D}$-driven pancreatic tumorigenesis.

Jag1 expression was up-regulated in ADM/PanIN lesions, and deletion of Jag1 accelerated ADM/PanIN formation in KJC mice, raising the possibility that Jag1 may suppress ADM initiation and progression to PanIN in the acinar compartment. On the other hand, abnormal ducts and PDAC displayed high level Jag1 expression, and deletion of Jag1 resulted in very low incidence of PDAC, suggesting that Jag1 may be required for advancement from PanIN to PDAC. The apparent dual role of Jag1 in pancreatic cancer may be dependent on the stage of cancer initiation and progression, or the cell-of-origin of the lesion. One of the limitations of this study is the use of *Pdx1-Cre* in the modeling of pancreatic cancer. *Pdx1-Cre* mediates expression of *Kras*$^{G12D}$ in all lineages of the pancreas starting from embryonic stage. Activation of oncogenic Kras at this stage does not mimic the real situation paralleling PDAC, an elderly disease in humans. In addition, deletion of Jag1 during organogenesis may have caused developmental defects of the pancreas (40), thereby setting up a precondition for Kras-induced pancreatic cancer initiation and progression. Future studies using inducible CreER systems in adult mice will be required for the delineation of Jag1 functions in the pathogenesis of acinar- or ductal-originated pancreatic cancer.

Lkb1 is mainly described as a tumor suppressor, whereas overall Jag1 is shown to be pro-tumorigenic here. Paradoxically, we showed that Jag1 positively regulates Lkb1 expression in pancreatic cells. As discussed above, deletion of Jag1 accelerated ADM/PanIN in KJC mice, suggesting a tumor-suppressive role of Jag1 in the early stage of tumorigenesis. Loss of Jag1 led to the development of cystic lesions, which appears to involve down-regulation of Lkb1. Thus, both Jag1 and Lkb1 may function as a tumor suppressor in the pathogenesis of pancreatic cystic neoplasm. Germline *LKB1* loss-of-function mutations are responsible for Peutz–Jeghers syndrome, a disease characterized by a predisposition to gastrointestinal neoplasms. Lkb1 deletion in the mouse pancreas caused serous cystadenomas, a tumor type associated with Peutz–Jeghers syndrome (32). Interestingly, Lkb1-deficient pancreas exhibits similar phenotype as the Jag1-deficient pancreas, except that the latter requires Kras$^{G12D}$ to induce cystic lesions. Moreover, deletion of Lkb1 in the liver resulted in bile duct paucity leading to cholestasis (41), similar to Alagille syndrome, an autosomal dominant disorder caused predominantly by mutations in *JAG1*. There has been mutual regulation of Lkb1 and Notch signaling in the biliary system (41). Thus, crosstalk between Jag1-mediated Notch signaling and Lkb1 appears to function in both pancreatic and biliary tracts.

IPMN lesions are thought to arise from the progenitor niche of the ductal epithelium. In this study, we used *Sox9-CreER*, which has been shown to label about 70% of pancreatic ductal cells in adult mice (42), to induce the deletion of Jag1 and expression of Kras$^{G12D}$ in adult pancreatic ducts. With the same *Sox9-CreER*, concurrent Lkb1 deletion and Kras activation in the ductal epithelium resulted in IPMN in adult mice (36), whereas deletion of Jag1 combined with Kras$^{G12D}$ expression in the ductal epithelium failed to induce cystic neoplasm. The negative result suggests that the phenotype seen with the *Pdx1-Cre* is likely due to Jag1 being deleted during development in the progenitor cells of the pancreas. However, we could not rule out the possibility of Jag1 playing a role in the cells of the ductal lineage. Additional insults including duct obstruction or mutations in other genes such as *p53* may be required to initiate neoplasia from ductal cells. Future studies using *Sox9-CreER*–mediated Jag1 deletion/ Kras$^{G12D}$ expression in conjunction with pancreatic duct ligation or *p53* deletion may determine whether Jag1 plays a role in ductal-originated pancreatic cancer. Finally, the results from this study suggest that SCN may have arisen from a progenitor in the acinar compartment, which should be tested through the deletion of Jag1 with lineage-specific inducible CreER system.

# Materials and Methods

### Mouse strains

Mouse strain *Jag1*$^{flox/flox}$ (43) was kindly provided by Dr Freddy Radtke (Ecole Polytechnique Fédérale de Lausanne). *Kras*$^{LSL-G12D}$, *Pdx1-Cre*, *Rosa*$^{LSL-YFP}$, and *Sox9-CreER* mouse strains were purchased from the Jackson Laboratory. Mice were interbred and maintained on a mixed background. All mouse experiments were performed in accordance with a protocol approved by the Institutional Animal Care and Use Committee of the University of Mississippi Medical Center.

### Histology, immunohistochemistry, and Western blot analysis

These techniques were performed as per standard protocols. Primary antibodies used for immunostaining were Cytokeratin 19 (ab52625, 1:200; Abcam), Vimentin (No. 5741, 1:100; Cell Signaling), Smooth muscle actin-$\alpha$ (ab5694, 1:200; Abcam), Ki67 (ab16667, 1:100; Abcam), Cleaved Caspase-3 (No. 9661, 1:100; Cell Signaling), Muc5AC (MA1-38223, 1:100; Thermo Fisher Scientific), ER$\alpha$ (sc-542, 1:200; Santa Cruz), PR (sc-538, 1:200; Santa Cruz), Jagged1 (sc-390177, 1:100; Santa Cruz), Notch2 (University of Iowa, C651.6DbHN, 1:200; DSHB), Lkb1 (sc-32245, 1:100; Santa Cruz), YFP (A-11122, 1:200; Invitrogen), and Sox9 (AB5535, 1:100; EMD Millipore). Antibodies for Western blot were as follows: Jag1 (sc-6011, 1:1,000; Santa Cruz), Sox9 (AB5535, 1:1,000; EMD Millipore) and $\beta$-Actin (sc-81178, 1:2,000; Santa Cruz).

### Quantitative RT-PCR

Total RNA was extracted with RNeasy Mini Kit (QIAGEN) and reverse-transcribed by iScript cDNA synthesis kit (Bio-Rad). PCR was performed using QuantiTect SYBR Green PCR Kits (QIAGEN) with BioRad CFX96 qPCR System. The experiment was performed in triplicate, and results were normalized with the expression level of Gapdh and

compared with the rest. **(J)** Overall survival in pancreatic cancer patients with *SOX9* expression lower than 1× SD below the mean, in comparison with the rest. ****$P <$ 0.0001 (*t* test). Scale bars: 12.5 $\mu$m.
Source data are available for this figure.

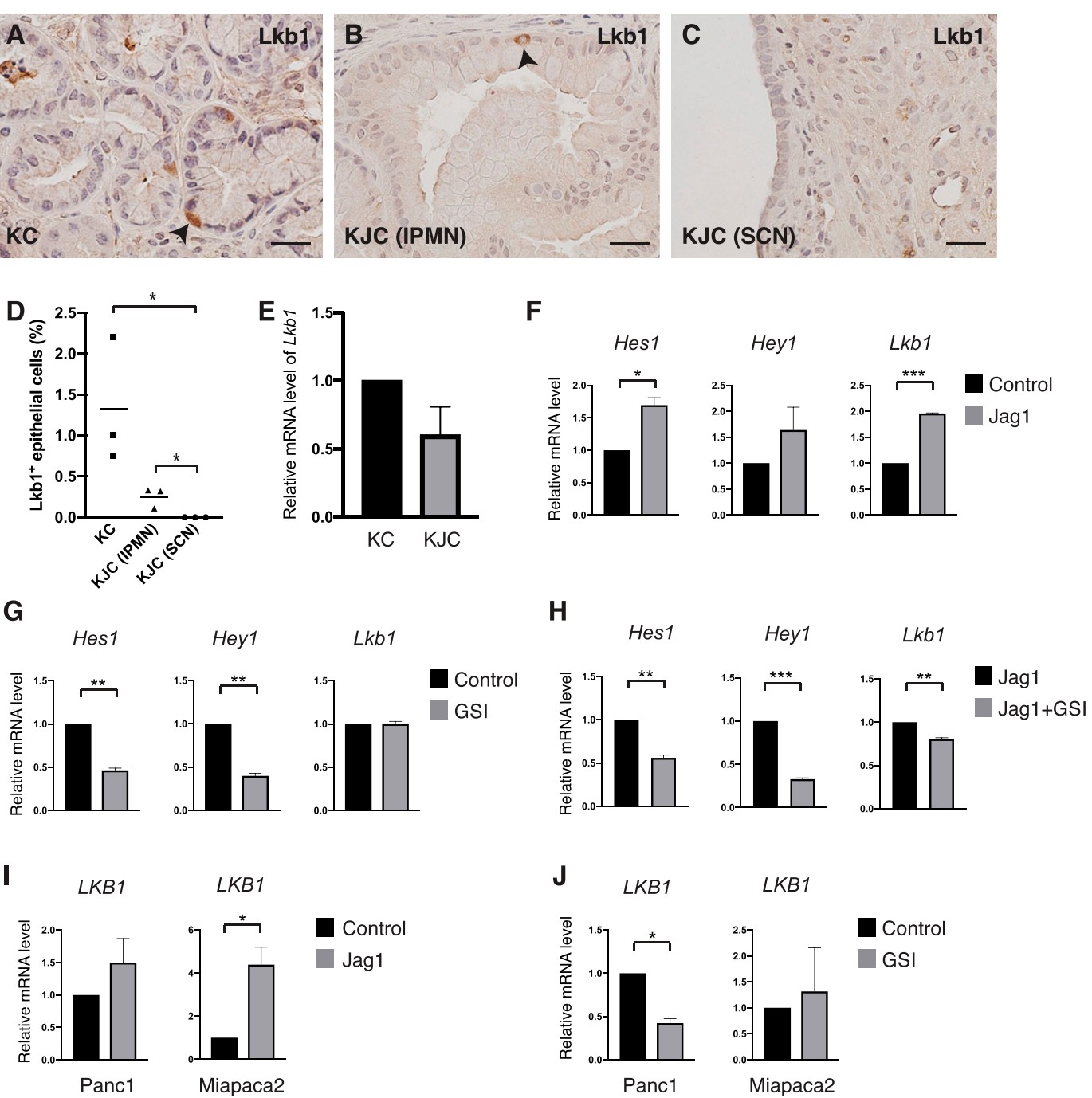

**Figure 6.    Regulation of Lkb1 expression by Jag1 in pancreatic cells.**
**(A, B, C)** Representative photomicrographs of immunostaining for Lkb1 in the pancreas of adult KC and KJC mice. **(A, B)** Arrows: Lkb1+ cells in PanIN lesions of KC mice (A) and intraductal papillary mucinous neoplasm–like lesions in KJC mice (B). **(D)** Quantification of Lkb1+ cells per total epithelial cells in KC pancreas and KJC pancreas with intraductal papillary mucinous neoplasm–like or serous cystic neoplasm–like lesions. **(E)** Relative mRNA level of *Lkb1* in the pancreas of KC and KJC mice determined by quantitative RT-PCR. Four mice per genotype were analyzed at 2–3 mo of age. **(F)** Relative mRNA levels of *Hes1*, *Hey1*, and *Lkb1* in KJC-derived pancreatic ductal cells grown on control or Jag1-precoated petri dish. **(G)** Relative mRNA levels of *Hes1*, *Hey1*, and *Lkb1* in KJC-derived pancreatic ductal cells treated with gamma secretase inhibitor (GSI) or vehicle control (DMSO). **(H)** Relative mRNA levels of *Hes1*, *Hey1*, and *Lkb1* in KJC-derived pancreatic cells cultured on Jag1-precoated petri dish in the presence of GSI or DMSO as control. **(I)** Relative mRNA levels of *LKB1* in human PDAC cell lines cultured in control or Jag1-precoated petri dish. **(J)** Relative mRNA level of *LKB1* in human PDAC cell lines treated with GSI or DMSO as control. *P < 0.05, **P < 0.01, ***P < 0.001 (*t* test). Scale bars: 12.5 μm.

presented as mean ± SEM. Primer sequences for mouse *Lkb1*, *Ctgf*, *Jag1*, *Hes1*, *Hes5*, *Hey1*, and *Hey2* have been described previously (7, 9, 44, 45).

### Cell culture

Miapaca2 and Panc1 were purchased from ATCC. Primary pancreatic cells were isolated from KJC mice with SCN-like lesions. Briefly, pancreatic tissue was rapidly collected, minced with blades, and plated on a 100-mm plate with DMEM and 10% FBS for 24 h. The supernatant was discarded and attached cells were allowed to grow until confluency. Cells were seeded into a six-well plate precoated with recombinant Jagged1-Fc chimera protein or recombinant IgG1 Fc protein as control (R&D Systems) at 10 $\mu$g/well in 100 $\mu$l PBS, or treated with GSI (MK-0752, 20 $\mu$M) or vehicle control (DMSO) for 48 h.

### Gene expression analysis of human data sets

Data sets used for *JAG1* expression analysis were downloaded from the International Cancer Genome Consortium (PDAC-AU, n = 188) and GEO (GSE28735, n = 90, and GSE16515, n = 32). Clinical follow-up data with overall survival information were only available for GSE28735(n = 42) and International Cancer Genome Consortium (PDAC-AU; n = 188), which were used to complete survival analysis. Expression data were normalized using Robust Multi-Array Average measure from R Bioconductor package "affy." Differential expression of *JAG1* in normal and tumor tissue was analyzed by one-way ANOVA. For the survival analysis, Kaplan–Meier survival curves were generated with the upper quartile as cutoff. The pancreatic adenocarcinoma data set (TCGA, Firehose Legacy; n = 186) hosted in cBioPortal (https://www.cbioportal.org) was used for the PathwayMapper analysis (24) and gene expression heat map with its online tool. Correlation and survival analysis related to *SOX9* expression was performed using the same data set and online tools in cBioPortal.

### Statistics

Statistical analyses were performed using Prism version 8.3.1 (GraphPad Software). The data are presented as the mean with SEM. Survival was analyzed using the Kaplan–Meier method and compared by the log-rank test. Correlations analysis was performed using chi-square test. Two-group comparisons were analyzed using two-tailed *t* test. *P*-value of 0.05 or less was considered statistically significant.

## Supplementary Information

## Acknowledgements

The authors thank Dr Freddy Radtke for providing *Jag1*<sup>flox</sup> mouse strain. This work was supported by the Intramural Research Support Program of the University of Mississippi Medical Center and by The Joe W and Dorothy Dorsett Brown Foundation to K Xu. We are grateful for the technical assistance from the University of Mississippi Medical Center Histology Core supported by the National Institutes of Health under Award Numbers P20GM104357 and P01HL51971.

## Author Contributions

W-C Chung: conceptualization, data curation, formal analysis, validation, investigation, visualization, methodology, and writing—original draft, review, and editing.
L Challagundla: data curation, formal analysis, validation, investigation, visualization, methodology, and writing—original draft, review, and editing.
Y Zhou: data curation, formal analysis, validation, investigation, visualization, methodology, and writing—original draft, review, and editing.
M Li: conceptualization, data curation, investigation, methodology, and writing—original draft, review, and editing.
A Atfi: conceptualization, formal analysis, validation, investigation, visualization, methodology, and writing—original draft, review, and editing.
K Xu: conceptualization, data curation, formal analysis, supervision, funding acquisition, validation, investigation, visualization, methodology, project administration, and writing—original draft, review, and editing.

## Conflict of Interest Statement

The authors declare that they have no conflict of interest.

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
