## [Reviewer comments · Life Science Alliance]

Life Science Alliance

Loss of Jag1 cooperates with oncogenic Kras to induce pancreatic cystic neoplasms

Wen-Cheng Chung, Lavanya Challagundla, Yunyun Zhou, Min Li, Azeddine Atfi, and Keli Xu
DOI: <https://doi.org/10.26508/lsa.201900503>

Corresponding author(s): Keli Xu, University of Mississippi Medical Center

Review Timeline:

Submission Date:	2019-07-26
Editorial Decision:	2019-08-15
Revision Received:	2020-10-28
Editorial Decision:	2020-11-17
Revision Received:	2020-11-18
Accepted:	2020-11-19

Scientific Editor: Shachi Bhatt

Transaction Report:

August 15, 2019

Re: Life Science Alliance manuscript #LSA-2019-00503-T

Dr. Keli Xu
University of Mississippi Medical Center
Cancer Institute
2500 North State Street
Jackson, MS 39216

Dear Dr. Xu,

Thank you for submitting your manuscript entitled "Loss of Jag1 cooperates with oncogenic Kras to induce pancreatic cystic neoplasms" to Life Science Alliance. The manuscript was assessed by expert reviewers, whose comments are appended to this letter.

As you will see, the reviewer have overlapping concerns and think that your claims are currently not supported by the data provided. Importantly, the imaging analyses are not convincing. Both reviewers provide constructive input to help you address the issues they noted and we would thus like to invite you to submit a revised version of your manuscript to us, addressing all concerns raised.

During the cross-commenting session the reviewers also added that you should verify your anti-Jag1 antibody before attempting to improve the stainings, as many commercialized antibodies don't work well on tissue sections. They further suggested to improve the Sox9 staining (figure 5) by using the anti-Sox9 from Millipore (AB5535), which works well in IHC and IF (citrate pH6, microwave retrieval).

Thank you for this interesting contribution to Life Science Alliance. We are looking forward to receiving your revised manuscript.

Sincerely,

B. MANUSCRIPT ORGANIZATION AND FORMATTING:

Reviewer #1 (Comments to the Authors (Required)):

In this paper, Chung et al. describe a tumor-promoting role for Jag1 in pancreas. Loss of Jag1 induces a switch to form SCN-like lesions rather than PDAC. Mechanistically, Jag1 mediates its effect by increasing the expression of Sox9 a master regulator of acinar-to-ductal metaplasia and neoplasia development.

The data are seriously interpreted and the paper fits into the scope of the journal. However, it cannot be published in its actual form and should undergo some essential modifications, before acceptance.

Major comments:

1- Validation of Jag1^{fl/fl} must be performed by western blot to confirm the deletion of Jag1 protein. RTqPCR of Jag1 is complementary but not sufficient.

2- Conclusion of the first paragraph "Jag1-mediated Notch signaling appears to suppress proliferation of exocrine cells" is not correct, as this aspect of cell proliferation was not assessed in Figure 1. At this stage, authors must provide a first mini-conclusion about the phenotypic changes that they observe using their KJC model. Please modify this sentence as appropriate.

3- I am very careful in the interpretation of results presented in Figure 4. Especially, Figure 4D and F. They do not show a real staining of Jag1. Figure 4D do not show any staining for Jag1 it is only high background due to high exposure time.

Figure 4F the staining for Jag1 is not really convincing as the structures remaining inside the lumen of PanIN lesions usually have non-specific staining with whatever antibody. IHC images are more convincing showing Jag1 staining in the cytoplasm. Immunofluorescence images must be removed and replaced with IHC, using a double staining for YFP and Jag1 (HRP and PA for example). If not possible, authors can show staining for YFP and Jag1 separately on adjacent tissue sections.

4- Figure 5E, Jag1 staining is not convincing. This figure must be removed and replaced by another one showing clear expression of Jag1. Otherwise, the conclusion of the authors about the ductal expression of Jag1 in WT mice is not supported.

5- The rationale that Jag1 or Lkb1 deficiency induce similar type of cystic lesions could be logical but still intriguing. Lkb1 is mainly described as a tumor suppressor, while Jag1 is shown to be pro-tumoral here. Could the authors explain why a tumor suppressor like Lkb1 is controlled by a pro-tumoral factor like Jag1?

6- It is preferable to provide western blot data showing increase in Sox9 expression at the protein level when cells are treated with Jag1. The result is interesting and should be corroborated.

7- The authors highlighted an important point by comparing the Pdx1-Cre model that initiates tumorigenesis process in all pancreatic progenitor cells and Sox9CreER model that expresses oncogenic Kras only in adult ductal cells. Usually, when we induce oncogenic Kras in acinar or ductal adult cells, additional insults like inflammation or mutations in other genes like p53 are necessary to initiate neoplasia. We can see that the biological responses are not the same and this is not surprising from my point of view. Authors should emphasize in their discussion the limitation of their Pdx1-Cre model. Despite the usefulness, that Pdx1-cre model has provided during the past years, it is not the ideal model to study the cell of origin of PDAC. Pancreatic cancer is an elderly disease and should be studied in adult mice using CreER systems, rather than Pdx1-cre. This point must be

discussed in more details at the end of the paper.

Minor comments:

- 1- YFP-positive stromal cells should be better indicated by arrows in Figure S2 C and F. Or another figure showing clearly stained stromal cells must be added.
- 2- Presentation of western blot quantification must be changed. An average decrease of sox9 must be presented instead of a point-by-point quantification. Statistical analysis should be also added.
- 3- It is preferable that authors state throughout the text if the difference is statistically significant or if it is just a tendency.
- 4- The statistical strategy used by the authors must be clearly defined in a separated subheading within the Material and Methods section.
- 5- In Figure 5L, E-cadherin is not a good loading control. Another marker like Actin or Hsc70 could be used.

Reviewer #2 (Comments to the Authors (Required)):

In the manuscript entitled "Loss of Jag1 cooperates with oncogenic Kras to induce pancreatic cystic neoplasms", Chung and colleagues show how Jag1 deletion accelerates Kras-induced ADM and PanIN formation, however, instead of progressing to PDAC, these lesions give rise to non-mucinous cysts (resembling SCN) and some IPMN. They propose that the phenotypic switch between PDAC and the benign cysts is mediated by Lkb1 and Sox9. It is an example of how finely tuned Notch signalling can act as a tumour suppressor in the Pancreas (by delaying lesion progression) while at the same time its imbalance can promote the appearance of benign lesions instead of progression into malignancy.

Result 1: Deletion of Jag1 accelerates Kras-driven ADM and PanIN formation

Data are strongly supportive.

Minor comments:

- Gene names in mRNA quantifications should be italicised in Figure 1

Result 2: A phenotypic switch from ductal adenocarcinoma to cystic neoplasms by Jag1 deletion

Data are supportive but I have some concerns. From the text it seems that out of {greater than or equal to}3 month old KJC mice, there are only 1 case of PDAC and 4 cases of ADM/PanIN, all the rest presenting SCN or IPMN. However, out of the "rest", the table shows that 9 of them indeed only present small cystic neoplasms but also PanIN - which is a bit misleading. Authors need to clarify on the text the presence of PanIN in these 9 other mice. Also, authors should perform additional experiments in KC mice and include a quantification of the lesions observed in these animals at least in the longer time-points. Expected time-frame: 6-8 months.

Supplemental figure 1 supporting this main point shows the ulcerating facial skin lesions that prompted euthanizing the animal, precluding the analysis of pancreas-related death. I think that this is a really interesting phenotype (mostly due to its penetrance), and I understand that it is out

of the scope of the study. However, I would expect a bit more information on as to why they might appear under a Pdx1-cre model - is Pdx1 also expressed on facial skin? Did these lesions appear somewhere else? Could they be related to a diabetic syndrome? How is the glucose metabolism affected on these mice?

Additional experiments: none, because I understand is out of the scope of the manuscript, but authors should add a bit more discussion about Pdx1 expression out of the pancreas.

Result 3: Characterization of cystic neoplasms in KJC mice

Data are strongly supportive.

Supplemental figure 2 does not show clear negativity of adipocytes. Additional experiments: improve YFP staining - it might be clearer in a less exposed picture or in an immunofluorescence. Also applies to the claim that some stromal cells appear YFP-positive, it seems that staining is overdeveloped. I would recommend adding a negative control of the IHC in the same type of tissue. Expected time frame: 1-2 months.

Result 4: Jag1 expression is lost in SCN but retained in IPMN and PDAC

IHC data are not supportive. No clear membrane localisation of Jag1 can be appreciated in any of the pictures. Additional experiments should be performed to:

1. Improve Jag1 staining and show some higher magnification images where membrane localisation can be clearly seen, as expected for a membrane-localised protein.
2. To confirm functionality of Jag1 expression, additional cleaved (active) Notch1 or Notch2 IHC would be necessary.

Expected time-frame: 1-2 months.

Human data only analysed with JAG1 mRNA levels. Authors should perform the additional analyses:

1. In the JAG1-high samples, is there a Jag1-Notch signature active?
2. Is there a correlation between JAG1 levels and LKB1 and/or SOX9? Do they also correlate with survival?

Expected time-frame: 1-2 months.

Minor comments:

- Italicise JAG1 in Figure 4 if it corresponds to mRNA levels.

Result 5: Jag1 regulates Lkb1 expression in pancreatic ductal cells

Data are supportive, but quantification of several pictures/fields and percentage of Lkb1+ cells per total epithelial cells counted would be advisable.

Additional experiments:

- Treat the primary cells plated over Jag1 with GSI to confirm the affirmation that "Lkb1 is regulated by Jag1-dependent Notch signalling"

Minor comments for Figure 5:

- Indicating the staining in each set of pictures would help (Lkb1 in Fig5 A-C and E-F; Sox9 in Fig5 I-K).
- Gene names in mRNA quantifications should be italicised.

Result 6: Loss of Sox9 expression in cystic neoplasms in KJC mice

Data are not supportive. IHC data are not supportive at all. Figure 5 I-K pictures do not show a "complete loss or drastic decrease in Sox9 expression in IPMN (J) and SCN-like (I) lesions compared to more advanced lesions (K)". Moreover, from the primary cell culture experiment shown in Fig 5M, suggesting a "direct regulation of Sox9 by Jag1" is too strong a statement. Is Notch signalling active

in these cells? Are there Rbpj binding sites in the Sox9 promoter?

Additional experiments:

Sox9 staining should be better shown as well as ICN1 status in the same lesions.

Result 7: Ductal cell-specific deletion of Jag1 in the adult pancreas does not lead to cystic neoplasms

In this last section the authors delete Jag1 in adult ductal cells using a Sox9-cre. The negative results suggest that the phenotype seen with the Pdx1-cre is due to Jag1 being deleted during development in the progenitor cells. IPMN lesions are thought to arise from the progenitor niche of the ductal epithelium - is Sox9 not expressed in the ductal progenitors? Or does Jag1 not play a role at all in the cells of the ductal lineage, but only if they come from acinar cells through ADM? These questions need to be addressed.

Reviewer #1

In this paper, Chung et al. describe a tumor-promoting role for Jag1 in pancreas. Loss of Jag1 induces a switch to form SCN-like lesions rather than PDAC. Mechanistically, Jag1 mediates its effect by increasing the expression of Sox9 a master regulator of acinar-to-ductal metaplasia and neoplasia development.

The data are seriously interpreted and the paper fits into the scope of the journal. However, it cannot be published in its actual form and should undergo some essential modifications, before acceptance.

We thank the Reviewer for their positive comments and careful review, which helped improve the manuscript. Below please find our response to comments as well as suggested changes in blue.

Major comments:

1- Validation of Jag1^{fl/fl} must be performed by western blot to confirm the deletion of Jag1 protein. RTqPCR of Jag1 is complementary but not sufficient.

Western blot analysis of Jag1 in the pancreas of Jag1^{fl/fl} and Jag1^{KO} mice is added in the revised Fig. 1B.

2- Conclusion of the first paragraph "Jag1-mediated Notch signaling appears to suppress proliferation of exocrine cells" is not correct, as this aspect of cell proliferation was not assessed in Figure 1. At this stage, authors must provide a first mini-conclusion about the phenotypic changes that they observe using their KJC model. Please modify this sentence as appropriate.

We thank the Reviewer for this comment. The mini-conclusion has been changed and now reads as follows: "Jag1-mediated Notch signaling appears to suppress ADM and PanIN formation caused by Kras^{G12D} expression in the pancreas starting from developmental stage". (Page 6)

3- I am very careful in the interpretation of results presented in Figure 4. Especially, Figure 4D and F. They do not show a real staining of Jag1. Figure 4D do not show any staining for Jag1 it is only high background due to high exposure time.

Figure 4F the staining for Jag1 is not really convincing as the structures remaining inside the lumen of PanIN lesions usually have non-specific staining with whatever antibody. IHC images are more convincing showing Jag1 staining in the cytoplasm. Immunofluorescence images must be removed and replaced with IHC, using a double staining for YFP and Jag1 (HRP and PA for example). If not possible, authors can show staining for YFP and Jag1 separately on adjacent tissue sections.

We thank the Reviewer for raising this issue. Indeed, IHC for Jag1 showed no cytoplasmic or membranous staining in the ductal cells of wild type mice or in the structures inside the lumen of PanIN lesions in KJC mice. Therefore, we have removed fluorescence staining of Jag1, and replaced with IHC of Jag1 (Fig. 4A-H).

At one month of age, Jag1 was undetectable by IHC in the ductal cells in wildtype mice (Fig. 4A). Jag1 was upregulated in a subset of ADM lesions (Fig. 4B) and abnormal ducts (Fig. 4C) in KJC mice, indicating that deletion of Jag1 was incomplete in these mice. By 2-3 months of age, many KJC mice had developed SCN-like lesions, where the ductal cells were completely negative for Jag1, while blood vessels in the same section stained positive for Jag1 (Fig. 4D, G). In contrast, high level Jag1 expression was detected in the ductal cells of a rarely formed PDAC in KJC mice (Fig. 4E, H).

4- Figure 5E, Jag1 staining is not convincing. This figure must be removed and replaced by another one showing clear expression of Jag1. Otherwise, the conclusion of the authors about the ductal expression of Jag1 in WT mice is not supported.

Figure 5E was Lkb1 staining (not Jag1 staining) in WT mice at postnatal day 20. We agree with the reviewer that this staining is not convincing. We repeated Lkb1 staining in WT mice and found no clear expression of Lkb1 in postnatal WT mice. Therefore, we have removed Figure 5E and F.

5- The rationale that Jag1 or Lkb1 deficiency induce similar type of cystic lesions could be logical but still intriguing. Lkb1 is mainly described as a tumor suppressor, while Jag1 is shown to be pro-tumoral here. Could the authors explain why a tumor suppressor like Lkb1 is controlled by a pro-tumoral factor like Jag1?

We appreciate Reviewer for this thoughtful comment. Although overall Jag1 is shown to be pro-tumoral in this study, deletion of Jag1 accelerated ADM and PanIN formation in KJC mice, suggesting a tumor-suppressive role for Jag1 in the early stage of tumor development. In addition, loss of Jag1 led to the development of cystic lesions, which appears to involve downregulation of Lkb1. Thus, both Jag1 and Lkb1 may function as a tumor suppressor in the pathogenesis of pancreatic cystic neoplasm. We have added this discussion on Page 16.

6- It is preferable to provide western blot data showing increase in Sox9 expression at the protein level when cells are treated with Jag1. The result is interesting and should be corroborated.

Per the reviewer's suggestion, we performed Western blot for Sox9 in KJC-derived primary cells treated with Jag1-Fc chimera protein or Fc protein as control. However, the results from different batches of primary cells were inconsistent. Without the corroboration at the protein level, we decided to remove the data showing increased Sox9 mRNA level in Jag1-treated cells.

Different batches of primary cells may represent distinct types of pancreatic lesions, or same type of lesion at different stages. The regulation of Sox9 expression by Jag1 may be dependent on the cell type from which lesions arise or the stage of lesion development. We showed that SCN-like and IMPN-like lesions stained negative or very weak for Sox9, whereas PDAC was strongly positive (Fig. 5A-C), and Sox9 protein level was significantly lower in KJC mice with cystic lesions compared to KC mice (Fig. 5G). Thus, deletion of Jag1 resulted in decreased Sox9 expression during Kras-initiated pancreatic tumor development. Interestingly, expressions of *SOX9* and *JAG1* are positively correlated in human pancreatic cancer patients (Fig. 5H). Whether Jag1 directly regulates Sox9 expression remains to be determined.

7- The authors highlighted an important point by comparing the Pdx1-Cre model that initiate tumorigenesis process in all pancreatic progenitor cells and Sox9CreER model that express oncogenic Kras only in adult ductal cells. Usually, when we induce oncogenic Kras in acinar or ductal adult cells, additional insults like inflammation or mutations in other genes like p53 are necessary to initiate neoplasia. We can see that the biological responses are not the same and this is not surprising from my point of view. Authors should emphasize in their discussion the limitation of their Pdx1-Cre model. Despite the usefulness, that Pdx1-cre model has provided during the past years, it is not the ideal model to study the cell of origin of PDAC. Pancreatic cancer is an elderly disease and should be studied in adult mice using CreER systems, rather than Pdx1-cre. This point must be discussed in more details at the end of the paper.

We thank the reviewer for the comment and valuable suggestion. We have added the following paragraph in the Discussion (Page 15-16):

“One of the limitations of this study is the use of *Pdx1-Cre* in the modeling of pancreatic cancer. *Pdx1-Cre* mediates expression of *Kras*^{G12D} in all lineages of the pancreas starting from embryonic stage. Activation of oncogenic Kras at this stage does not mimic the real situation paralleling PDAC, an elderly disease in humans. In addition, deletion of Jag1 during organogenesis may have caused developmental defects of the pancreas, thereby setting up a precondition for Kras-induced pancreatic cancer initiation and progression. Future studies using inducible CreER systems in adult mice will be required for the delineation of Jag1 functions in the pathogenesis of acinar- or ductal-originated pancreatic cancer.”

Minor comments:

1- YFP-positive stromal cells should be better indicated by arrows in Figure S2 C and F. Or another figure showing clearly stained stromal cells must be added.

A new image showing clearly stained stromal cells has been added (Supplemental Fig. S2 F).

2- Presentation of western blot quantification must be changed. An average decrease of sox9 must be presented instead of a point-by-point quantification. Statistical analysis should be also added. This has been changed per the reviewer's suggestion (Fig. 5G). The density of the Sox9 band relative to the loading control (β -actin) was measured in each sample. An average decrease of Sox9 level in KJC mice compared to KC mice (n=3) is presented ($P < 0.01$, Student's t-test).

3- It is preferable that authors state throughout the text if the difference is statistically significant or if it is just a tendency.

The authors thank the reviewer for this comment. We have performed statistical analysis for each comparison and stated throughout the text if the difference is statistically significant or it is just a tendency.

4- The statistical strategy used by the authors must be clearly defined in a separated subheading within the Material and Methods section.

This has been added in the Material and Methods section.

5- In Figure 5L, E-cadherin is not a good loading control. Another marker like Actin or Hsc70 could be used.

We replaced E-cadherin with β -actin as the loading control (Fig. 5G).

Reviewer #2

In the manuscript entitled "Loss of Jag1 cooperates with oncogenic Kras to induce pancreatic cystic neoplasms", Chung and colleagues show how Jag1 deletion accelerates Kras-induced ADM and PanIN formation, however, instead of progressing to PDAC, these lesions give rise to non-mucinous cysts (resembling SCN) and some IPMN. They propose that the phenotypic switch between PDAC and the benign cysts is mediated by Lkb1 and Sox9. It is an example of how finely tuned Notch signalling can act as a tumour suppressor in the Pancreas (by delaying lesion progression) while at the same time its imbalance can promote the appearance of benign lesions instead of progression into malignancy.

We thank the Reviewer for their thorough review and constructive suggestions. Below please find our response to the comments and suggested changes in blue.

Result 1: Deletion of Jag1 accelerates Kras-driven ADM and PanIN formation

Data are strongly supportive.

Minor comments:

- Gene names in mRNA quantifications should be italicised in Figure 1

This has been corrected.

Result 2: A phenotypic switch from ductal adenocarcinoma to cystic neoplasms by Jag1 deletion

Data are supportive but I have some concerns. From the text it seems that out of (greater than or equal to) 3 month old KJC mice, there are only 1 case of PDAC and 4 cases of ADM/PanIN, all the rest presenting SCN or IPMN. However, out of the "rest", the table shows that 9 of them indeed only present small cystic neoplasms but also PanIN - which is a bit misleading. Authors need to clarify on the text the presence of PanIN in these 9 other mice. Also, authors should perform additional experiments in KC mice and include a quantification of the lesions observed in these animals at least in the longer time-points. Expected time-frame: 6-8 months.

We thank the Reviewer for this comment. In the revised manuscript, the presence of PanIN in these 9 other mice has been clarified, and quantification of the lesions in KC animals is presented for comparison with KJC mice. The text on Page 7 now reads as follows:

“Histological examination of the pancreas in 28 KJC mice (≥ 3 months) found 9 cases of small cystic neoplasm (coexisted with PanIN), 14 cases (50%) of SCN or IPMN lesions, and only one case (3.6%) of PDAC (Fig. 2K, L). For comparison, there were 2 cases (6.0%) of IPMN-like lesion (no SCN-like lesion) and up to 11 cases (33%) of PDAC out of 33 KC mice older than 3 months (Fig. 2O and Table 1). Thus, deletion of *Jag1* was associated with significantly increased incidence of cystic neoplasms and decreased incidence of PDAC (χ^2 test, $p < 0.0001$), suggesting a switch from ductal adenocarcinoma to cystic neoplasms.”

Supplemental figure 1 supporting this main point shows the ulcerating facial skin lesions that prompted euthanizing the animal, precluding the analysis of pancreas-related death. I think that this is a really interesting phenotype (mostly due to its penetrance), and I understand that it is out of the scope of the study. However, I would expect a bit more information on as to why they might appear under a *Pdx1-cre* model - is *Pdx1* also expressed on facial skin? Did these lesions appear somewhere else? Could they be related to a diabetic syndrome? How is the glucose metabolism affected on these mice?

Additional experiments: none, because I understand is out of the scope of the manuscript, but authors should add a bit more discussion about *Pdx1* expression out of the pancreas.

We appreciate the Reviewer’s insightful comment. The skin lesions of KJC mice were predominantly on the face, with a few near the anus. It has been shown that *Pdx1* is physiologically expressed in the adult mouse epidermis, and in vitro analysis revealed differentiation-dependent expression of *Pdx1* in terminally differentiated keratinocytes [1]. Although we cannot rule out the possibility of the skin phenotype being related to a diabetic syndrome, the fully penetrated skin phenotype in KJC mice suggests that loss of *Jag1*-mediated Notch signaling may cooperate with oncogenic *Kras* to induce skin carcinogenesis. Interestingly, deletion of *Notch1* also increased susceptibility to *Kras*^{G12D}-induced skin carcinogenesis with *Pdx1-Cre* [1]. The above discussion has been added in the revised manuscript (Page 7).

Result 3: Characterization of cystic neoplasms in KJC mice

Data are strongly supportive.

Supplemental figure 2 does not show clear negativity of adipocytes. Additional experiments: improve YFP staining - it might be clearer in a less exposed picture or in an immunofluorescence. Also applies to the claim that some stromal cells appear YFP-positive, it seems that staining is overdeveloped. I would recommend adding a negative control of the IHC in the same type of tissue. Expected time frame: 1-2 months.

We performed IHC of YFP with a negative control as per the Reviewer’s suggestion. The new images show clear negativity of adipocytes (arrow in Supplemental Fig. S2 E) and positivity in a small subset of stromal cells (arrows in Supplemental Fig. S2 F). The negative control of the IHC is shown in Supplemental Fig. S2 D.

Result 4: *Jag1* expression is lost in SCN but retained in IPMN and PDAC

IHC data are not supportive. No clear membrane localisation of *Jag1* can be appreciated in any of the pictures. Additional experiments should be performed to:

1. Improve *Jag1* staining and show some higher magnification images where membrane localisation can be clearly seen, as expected for a membrane-localised protein.

We improved *Jag1* immunostaining with a different antibody. As shown in the revised Fig. 4, ductal cells lining the SCN-like lesion were completely negative for *Jag1*, whereas blood vessels on the same section stained positive for *Jag1* (Fig. 4D, G). To the contrary, a rarely formed PDAC contained ductal cells with clear cytoplasmic and some membrane staining of *Jag1* (Fig. 4E, H).

2. To confirm functionality of Jag1 expression, additional cleaved (active) Notch1 or Notch2 IHC would be necessary.

Expected time-frame: 1-2 months.

We thank the Reviewer for this constructive suggestion. We have performed IHC for Jag1 and cleaved Notch1 on consecutive sections. There was no overlapping between Jag1 expression and presence of cleaved Notch1 (data not shown). Then we performed IHC for Jag1 and Notch2 on consecutive sections. Cytoplasmic staining of Jag1 and nuclear staining of Notch2 were observed at the same location on consecutive sections, suggesting activation of Notch2 by Jag1 (Fig. 4F, I). Interestingly, Notch2 has been shown to function in ductal cells and is required for PanIN progression and malignant transformation, and deletion of *Notch2* combined with *Kras*^{G12D} expression led to MCN-like cystic lesions in a subset of mice [2]. These findings suggest that Jag1 may regulate PanIN progression through Notch2 signaling.

Human data only analysed with JAG1 mRNA levels. Authors should perform the additional analyses:

Expected time-frame: 1-2 months.

1. In the JAG1-high samples, is there a Jag1-Notch signature active?

We analyzed *JAG1* expression in The Cancer Genome Atlas (TCGA) pancreatic adenocarcinoma data set hosted in the cBioPortal database. PathwayMapper [3] analysis found six of the Notch signaling pathway genes, including *JAG1*, *JAG2*, *NOTCH3*, *MAML1*, *MAML2* and *MAML3*, were upregulated in this data set (Fig. 4L). Expression heatmap of these genes showed an active Jag1-Notch signature in the JAG1-high samples (Fig. 4M).

2. Is there a correlation between JAG1 levels and LKB1 and/or SOX9? Do they also correlate with survival?

Analysis using TCGA pancreatic adenocarcinoma data set hosted in the cBioPortal database found a positive correlation between mRNA levels of *JAG1* and *SOX9* (Fig. 5H), however, a negative correlation between *JAG1* and *LKB1* (data not shown). Although we showed that Jag1 positively regulates Lkb1, Jag1 is also highly expressed in the vasculature (Fig. 4G). The negative correlation between *JAG1* and *LKB1* expressions may in part be due to high vascularity in advanced tumors.

Similar to *JAG1*, high expression of *SOX9* is associated with poor overall survival, whereas low *SOX9* expression is associated with significantly longer survival (Fig. 5I, J). In contrast to *JAG1*, low expression of *LKB1* is associated with poor overall survival (data not shown). Jag1 may act as a tumor suppressor in delaying ADM/PanIN development and preventing cystic lesions. However, Jag1 is overall to be pro-tumoral (deletion of Jag1 led to the development of largely benign cystic lesion instead of malignant carcinoma), whereas Lkb1 is mainly described as a tumor suppressor [4]. This may explain why low *LKB1* expression is related to poor survival whereas low *JAG1* is associated with longer survival.

Minor comments:

- Italicise JAG1 in Figure 4 if it corresponds to mRNA levels.

JAG1 in Figure 4 corresponds to mRNA levels. It is now italicized.

Result 5: Jag1 regulates Lkb1 expression in pancreatic ductal cells

Data are supportive, but quantification of several pictures/fields and percentage of Lkb1+ cells per total epithelial cells counted would be advisable.

Quantification of Lkb1⁺ epithelial cells in the KC and KJC pancreas (with IPMN-like or SCN-like lesion) has been added in Fig. 6D.

Additional experiments:

- Treat the primary cells plated over Jag1 with GSI to confirm the affirmation that "Lkb1 is regulated by Jag1-dependent Notch signalling"

We appreciate the suggestion of this experiment. Primary cells from the KJC pancreas were plated over Jag1 with GSI (or DMSO as control). The presence of GSI resulted in decreased mRNA levels of *Hes1* and *Hey1*, as well as a modest but significant decrease in the mRNA level of *Lkb1*, supporting that *Lkb1*

expression is regulated by Jag1-dependent Notch signaling (Fig. 6H).

Minor comments for Figure 5:

- Indicating the staining in each set of pictures would help (Lkb1 in Fig5 A-C and E-F; Sox9 in Fig5 I-K).

The staining in each set of pictures has been indicated in revised Fig. 5A-F and Fig. 6A-C.

- Gene names in mRNA quantifications should be italicised.

This has been corrected.

Result 6: Loss of Sox9 expression in cystic neoplasms in KJC mice

Data are not supportive. IHC data are not supportive at all. Figure 5 I-K pictures do not show a "complete loss or drastic decrease in Sox9 expression in IPMN (J) and SCN-like (I) lesions compared to more advanced lesions (K)". Moreover, from the primary cell culture experiment shown in Fig 5M, suggesting a "direct regulation of Sox9 by Jag1" is too strong a statement. Is Notch signalling active in these cells? Are there Rbpj binding sites in the Sox9 promoter?

Additional experiments:

Sox9 staining should better shown as well as ICN1 status in the same lesions.

We agree with Reviewer that the Sox9 staining was not supportive. We have improved the IHC with a new anti-Sox9 antibody. As shown in revised Fig. 5A-C, ductal cells lining the SCN-like lesion were completely negative for Sox9, IPMN-like lesions showed no or very weak Sox9 staining, whereas a rarely formed PDAC showed Sox9 nuclear staining in the ductal cells. IHC for Notch2 detected nuclear staining in PDAC cells, but not in the SCN-like or IPMN-like lesions (Fig. 5D-F). These results suggest that cystic neoplasms in KJC mice have lost Sox9 expression, accompanied by loss of Notch2 activation.

Although there is a putative Rbpj binding site in the *Sox9* promoter region, we agree with the reviewer that suggesting a "direct regulation of Sox9 by Jag1" is too strong a statement. Reviewer #1 also suggested performing Western blot for Sox9 to corroborate the RT-PCR results. In fact, Western blot analysis using different batches of primary cells had inconsistent results. Without the corroboration at the protein level, we decided to remove the RT-PCR data showing increased Sox9 expression in Jag1-treated cells.

Given that SCN-like and IPMN-like lesions stained negative or very weak for Sox9, whereas PDAC was strongly positive (Fig. 5A-C), and Sox9 protein level was significantly lower in KJC mice with cystic lesions compared to KC mice (Fig. 5G), it appears that deletion of Jag1 resulted in decreased Sox9 expression during Kras-initiated pancreatic tumor development. Interestingly, expressions of *SOX9* and *JAG1* are positively correlated in human pancreatic cancer patients (Fig. 5H). Whether Jag1 directly regulates Sox9 expression remains to be determined.

Result 7: Ductal cell-specific deletion of Jag1 in the adult pancreas does not lead to cystic neoplasms

In this last section the authors delete Jag1 in adult ductal cells using a Sox9-cre. The negative results suggest that the phenotype seen with the Pdx1-cre is due to Jag1 being deleted during development in the progenitor cells. IPMN lesions are thought to arise from the progenitor niche of the ductal epithelium - is Sox9 not expressed in the ductal progenitors? Or does Jag1 not play a role at all in the cells of the ductal lineage, but only if they come from acinar cells through ADM? These questions need to be addressed.

We used a *Sox9-CreER* strain that has been shown to label 70% of pancreatic ductal cells in the adult mice [5]. With this *Sox9-CreER*, combined Lkb1 deletion and Kras^{G12D} activation in the ductal epithelium was able to induce IPMN in adult mice [6], suggesting that Sox9 is expressed in the ductal progenitors.

However, we could not rule out the possibility of Jag1 playing a role in the cells of the ductal lineage.

Additional insults including duct obstruction or mutations in other genes may be required to initiate neoplasia from ductal cells. Future studies using *Sox9-CreER*-directed Jag1 deletion/Kras^{G12D} expression in conjunction with p53 deletion or pancreatic duct ligation may determine whether Jag1 plays a role in

ductal-originated pancreatic cancer. This discussion has been added at the end of the manuscript (Page 17).

References:

1. Mazur PK, Gruner BM, Nakhai H, Sipos B, Zimmer-Strobl U, Strobl LJ, Radtke F, Schmid RM, Siveke JT (2010) Identification of epidermal Pdx1 expression discloses different roles of Notch1 and Notch2 in murine Kras(G12D)-induced skin carcinogenesis in vivo. *PLoS One* **5**: e13578
2. Mazur PK, Einwachter H, Lee M, Sipos B, Nakhai H, Rad R, Zimmer-Strobl U, Strobl LJ, Radtke F, Kloppel G, *et al.* (2010) Notch2 is required for progression of pancreatic intraepithelial neoplasia and development of pancreatic ductal adenocarcinoma. *Proc Natl Acad Sci U S A* **107**: 13438-43
3. Bahceci I, Dogrusoz U, La KC, Babur O, Gao J, Schultz N (2017) PathwayMapper: a collaborative visual web editor for cancer pathways and genomic data. *Bioinformatics* **33**: 2238-2240
4. Morton JP, Jamieson NB, Karim SA, Athineos D, Ridgway RA, Nixon C, McKay CJ, Carter R, Brunton VG, Frame MC, *et al.* (2010) LKB1 haploinsufficiency cooperates with Kras to promote pancreatic cancer through suppression of p21-dependent growth arrest. *Gastroenterology* **139**: 586-97, 597 e1-6
5. Kopp JL, Dubois CL, Schaffer AE, Hao E, Shih HP, Seymour PA, Ma J, Sander M (2011) Sox9+ ductal cells are multipotent progenitors throughout development but do not produce new endocrine cells in the normal or injured adult pancreas. *Development* **138**: 653-65
6. Collet L, Ghurburrun E, Meyers N, Assi M, Pirlot B, Leclercq IA, Couvelard A, Komuta M, Cros J, Demetter P, *et al.* (2020) Kras and Lkb1 mutations synergistically induce intraductal papillary mucinous neoplasm derived from pancreatic duct cells. *Gut* **69**: 704-714

November 17, 2020

RE: Life Science Alliance Manuscript #LSA-2019-00503-TR

Dr. Keli Xu
University of Mississippi Medical Center
Cancer Institute
2500 North State Street
Jackson, Mississippi 39216

Dear Dr. Xu,

Thank you for submitting your revised manuscript entitled "Loss of Jag1 cooperates with oncogenic Kras to induce pancreatic cystic neoplasms". We would be happy to publish your paper in Life Science Alliance pending final revisions necessary to meet our formatting guidelines.

Along with the points listed below, please also attend to the following,

- please add your supplementary figure legends to the main manuscript text, directly under your main figure legends
- please upload your Table as an editable doc or excel file format and add your Table legend to the main manuscript text
- please add a callout for Figure S2 A, D, E to your main manuscript text
- please expand on the what the panels in Figure S1B-E show in the S1 Figure legend
- please re-word the Figure legends for Figure 3, 4 and S2 such that the panels are introduced in an alphabetical order

A. FINAL FILES:

-- High-resolution figure, supplementary figure and video files uploaded as individual files: See our detailed guidelines for preparing your production-ready images, <https://www.life-science->

alliance.org/authors

B. MANUSCRIPT ORGANIZATION AND FORMATTING:

Sincerely,

Shachi Bhatt, Ph.D.
Executive Editor
Life Science Alliance
<https://www.lsjournal.org/>
Tweet @SciBhatt @LSAJournal

Reviewer #1 (Comments to the Authors (Required)):

Authors have addressed all raised issues. The quality of the manuscript has been significantly improved. I recommend to accept the paper in its current form.

Reviewer #2 (Comments to the Authors (Required)):

In the revised manuscript entitled "Loss of Jag1 cooperates with oncogenic Kras to induce pancreatic cystic neoplasms", Chung and colleagues show how Jag1 deletion accelerates Kras-induced ADM and PanIN formation, however, instead of progressing to PDAC, these lesions give rise to non-mucinous cysts (resembling SCN) and some IPMN. They propose that the phenotypic switch between PDAC and the benign cysts is mediated by Lkb1 and Sox9. The authors have improved Yfp and Jag1 stainings and now show clear activation of Notch2. Since the Sox9 regulation at the protein level was unclear, they have removed the inconsistent results and now point out that further experiments should be performed to delve into how Jag1 directly or indirectly regulates Sox9 expression. The authors show more consistent data pointing towards Jag1 regulating Lkb1 expression through Notch, as this is precluded by the presence of GSI in the culture media. I thank the authors for performing these informative experiments and improvement of the pictures overall. They have also included discussion points raised by both reviewers, adding value to the manuscript. Overall, I think that all the data shown now are supportive of the main points of the paper and I have no further minor comments.

November 19, 2020

RE: Life Science Alliance Manuscript #LSA-2019-00503-TRR

Dr. Keli Xu
University of Mississippi Medical Center
Cancer Institute
2500 North State Street
Jackson, Mississippi 39216

Dear Dr. Xu,

Thank you for submitting your Research Article entitled "Loss of Jag1 cooperates with oncogenic Kras to induce pancreatic cystic neoplasms". It is a pleasure to let you know that your manuscript is now accepted for publication in Life Science Alliance. Congratulations on this interesting work.

DISTRIBUTION OF MATERIALS:

Again, congratulations on a very nice paper. I hope you found the review process to be constructive and are pleased with how the manuscript was handled editorially. We look forward to future exciting submissions from your lab.

Sincerely,

Shachi Bhatt, Ph.D.
Executive Editor
Life Science Alliance
<https://www.lsjournal.org/>
Tweet @SciBhatt @LSAJournal